



# Global emissions pathways under different socioeconomic scenarios for use in CMIP6: a dataset of harmonized emissions trajectories through the end of the century

Matthew J. Gidden[1], Keywan Riahi[1], Steven J. Smith[2], Shinichiro Fujimori[3], Gunnar Luderer[4], Elmar Kriegler[4], Detlef P. van Vuuren[5], Maarten van den Berg[5], Leyang Feng[2], David Klein[4], Katherine Calvin[2], Jonathan C. Doelman[5], Stefan Frank[1], Oliver Fricko[1], Mathijs Harmsen[5], Tomoko Hasegawa[3], Petr Havlik[1], Jérôme Hilaire[4,6], Rachel Hoesly[2], Jill Horing[2], Alexander Popp[4], Elke Stehfest[5], and Kiyoshi Takahashi[3]

[1]International Institute for Applied Systems Analysis, Schlossplatz 1, A-2361 Laxenburg, Austria
[2]Joint Global Change Research Institute, 5825 University Research Court, Suite 3500, College Park, MD 20740
[3]Center for Social and Environmental Systems Research, National Institute for Environmental Studies, 16-2 Onogawa, Tsukuba, Ibaraki 305-8506, Japan
[4]Potsdam Institute for Climate Impact Research, Member of the Leibniz Association, P.O. Box 60 12 03, D-14412 Potsdam, Germany
[5]PBL Netherlands Environmental Assessment Agency, Postbus 30314, 2500 GH The Hague, Netherlands
[6]Mercator Research Institute on Global Commons and Climate Change (MCC) gGmbH, EUREF Campus 19, Torgauer Str. 12-15, 10829 Berlin

**Correspondence:** Matthew J. Gidden (gidden@iiasa.ac.at)

**Abstract.** We present a suite of nine scenarios of future emissions trajectories of anthropogenic sources, a key deliverable of the ScenarioMIP experiment within CMIP6. Integrated Assessment Model results for 14 different emissions species and 13 emissions sectors are provided for each scenario with consistent transitions from the historical data used in CMIP6 to future trajectories using automated harmonization before being downscaled to provide higher emission source spatial detail. We find

5   that the scenarios span a wide range of end-of-century radiative forcing values, thus making this set of scenarios ideal for exploring a variety of warming pathways. The set of scenarios are bounded on the low end by a $1.9\,\mathrm{W\,m^{-2}}$ scenario, ideal for analyzing a world with end-of-century temperatures well below 2°C, and on the high-end by a $8.5\,\mathrm{W\,m^{-2}}$ scenario, resulting in an increase in warming of nearly 5°C over pre-industrial levels. Between these two extremes, scenarios are provided such that differences between forcing outcomes provide statistically significant regional temperature outcomes to maximize their

10   usefulness for downstream experiments within CMIP6. A wide range of scenario data products are provided for the CMIP6 scientific community including global, regional, and gridded emissions datasets.



# 1 Introduction

Scenario development and analysis play a crucial role in linking socioeconomic and technical progress to potential future climate outcomes by providing future trajectories of various emissions species including greenhouse gases, aerosols, and their precursors. These assessments and associated datasets allow for wide-ranging climate analyses including pathways of future
warming, localized effects of pollution emissions, and impacts studies, among others. By spanning a wide range of possible futures, including varied levels of emissions mitigation, pollution control, and socioeconomic development, scenarios provide a large multivariate space of potential near, medium, and long-term outcomes for study by the broader scientific community.

The result of scenario exercises have been used widely by national and international assessment bodies and the global scientific community. They have informed previous Assessment Reports by the Intergovernmental Panel on Climate Change
(Solomon et al., 2007; Stocker et al., 2013) as well as reports on more topical issues including the Special Report on Emissions Scenarios (SRES) (Nakićenović et al., 2000). The SRES scenarios were used extensively in the 3rd Phase of the Coupled Model Intercomparison Project (CMIP3)(Solomon et al., 2007), whereas the following generation of scenarios denoted the "Representative Concentration Pathways" (RCPs) were used to generate emissions trajectories in CMIP5 (Moss et al., 2010; van Vuuren et al., 2011; Taylor et al., 2012).
As initially described in Moss et al. (2010), a new framework has been utilized to design scenarios that combine socioeconomic and technological development, named the Shared Socioeconomic Pathways (SSPs), with future climate radiative forcing (RF) outcomes (RCPs) in a scenario matrix architecture (O'Neill et al., 2013; Kriegler et al., 2014; van Vuuren et al., 2013). This new structure provides two critical elements to the scenario design space: first, it standardizes all socioeconomic assumptions (e.g., population, GDP, and poverty, among others) across modeled representations of each scenario; second, it
allows for more nuanced investigation of the variety of pathways by which climate outcomes can be reached. Five different SSPs exist, with model quantifications that span potential futures of green or fossil-fueled growth (SSP1 (van Vuuren et al., 2017) and SSP5 (Kriegler et al., 2017)), high inequality between or within countries (SSP3 (Fujimori et al., 2017) and SSP4 (Calvin et al., 2017)), and a "middle of the road" scenario (SSP2 (Fricko et al., 2017)). For each SSP, a number of different RF targets can be met depending on policies implemented, either locally or globally, over the course of the century (Riahi et al.,
25  2017).

Scenarios provide critical input for climate models through their description and quantification of both land-use change as well as emissions trajectories. Of the total population of newly available scenarios produced with Integrated Assessment Models (IAMs), nine have been chosen for inclusion for study in ScenarioMIP, one of the dedicated CMIP6-endorsed MIPs (Eyring et al., 2016). The selection of scenarios is designed to allow investigation of two primary scientific questions: "How
does the Earth system respond to climate forcing?" and "How can we assess future climate changes given climate variability… and uncertainties in scenarios?" (O'Neill et al., 2016). In order to support an experimental design that can address these fundamental questions, scenarios where chosen that explore a wide range of future climate forcing that both complement and expand on prior work in CMIP5. While a given forcing pathway could be met with potentially many different SSPs, a specific SSP is chosen for each pathway according to three governing principles: "[maximizing] facilitation of climate research,



minimizing differences in climate between outcomes produced by the [chosen] SSP, and ensuring consistency with scenarios that are most relevant to the IAM and Impacts, Adaptation, and Vulnerability (IAV) communities" (O'Neill et al., 2016, p. 3469).

Selected scenarios sample a range of forcing outcomes (1.9-8.5 $\mathrm{W\,m^{-2}}$, calculated with the simple climate model MAGICC6 (Meinshausen et al., 2011a)), with sufficient spacing between forcing outcomes to provide statistically significant regional temperature outcomes (Tebaldi et al., 2015; O'Neill et al., 2016). The nine selected scenarios can be divided into two groups: four scenarios update the RCPs studied in CMIP5, achieving forcing levels of 2.6, 4.5, 6.0, and 8.5 $\mathrm{W\,m^{-2}}$, whereas five scenarios fill gaps not previously studied in the RCPs, including, a lower-bound 1.9 $\mathrm{W\,m^{-2}}$ scenario (Rogelj et al., 2018) corresponding to the most optimistic interpretation of Article 2 of the Paris Agreement (United Nations, 2016). Additionally, a new 'overshoot' scenario is included in the Tier 2 set in which forcing peaks and then declines to 3.4 $\mathrm{W\,m^{-2}}$ by 2100 in order to assess the climatic outcomes of such a pathway.

In order to provide historically consistent and spatially detailed emissions datasets for other scientists collaborating in CMIP6, scenario results are processed using methods of harmonization and downscaling, respectively. Harmonization refers to the alignment of model results with a common historical dataset. Historical data consistency is paramount for use in climate models which perform both historic and future runs, for which there must be smooth transitions between the two sets of emissions trajectories. Harmonization has been applied in previous studies (e.g., in SRES (Nakićenović et al., 2000) and the RCPs (van Vuuren et al., 2011; Meinshausen et al., 2011b)); however, systematic harmonization for which common rules and algorithms are applied across all models has not heretofore been performed (Rogelj et al., 2011). We harmonize emissions trajectories, therefore, with a newly-available methodology and software (*aneris*) (Gidden, 2017; Gidden et al., 2018) in order to address this need. We further downscale these results from their native model region spatial dimension to individual countries using techniques which take into account current and future emissions levels as well as socioeconomic progress (van Vuuren et al., 2007). An overview of the scenario selection and processing steps that comprise this study as well as its contributions to the broader CMIP6 community is shown in Figure 1.

The remainder of the paper is as follows. First, we discuss scenario selection, historical data aggregation, harmonization, and downscaling methods in Section 2. We then present harmonized model results, focusing on overall emissions trajectories, climate response outcomes, and the spatial distribution of key emissions species in Section 3. Finally, in Section 4, we discuss conclusions drawn from this study as well as guidelines for using the results presented herein in further CMIP6 experiments.

## 2 Data and Methods

### 2.1 Socioeconomic and Climate Scenarios

The global IAM community has developed a family of scenarios that describe a variety of possible socioeconomic futures (the SSPs). The formation, qualitative, and quantitative aspects of these scenarios have been discussed widely in the literature (O'Neill et al., 2017; KC and Lutz, 2014; Dellink et al., 2015; Jiang and O'Neill, 2015). We briefly summarize here relevant



**Figure 1.** The role of ScenarioMIP in the CMIP6 ecosystem. From a population of over 40 possible SSPs, nine are downselected in order to span the climatic and social dimensions of the ScenarioMIP SSP-RCP Matrix. Emissions trajectories developed from these scenarios then undergo harmonization to a common and consistent historical dataset, downscaling, and gridding. The resulting emissions datasets are then provided to the CMIP6 scientific community, in conjunction with future scenarios of land use (Hurtt, 2018), concentrations (Meinshausen, 2018), and other domain-specific datasets (e.g., VOC speciation and ozone concentrations).





narratives of the baseline SSPs concerning socioeconomic development (see, e.g., Figure A1), energy systems (Bauer et al., 2017), land use (Popp et al., 2017), Greenhouse Gas (GHG) emissions (Riahi et al., 2017), and air pollution (Rao et al., 2017).

SSPs 1 and 5 describe worlds with strong economic growth via sustainable and fossil-fuel pathways, respectively. In both scenarios, incomes increase substantially across the globe and inequality within and between countries is greatly reduced;

however, this growth comes at the expense of potentially large impacts from climate change in the case of SSP5. Demand for energy and resource intensive agricultural commodities such as ruminant meat is significantly lower in SSP1 due to changes in behavior and advances in energy efficiency. In both scenarios, pollution controls are expanded in high-income economies with other nations catching up relatively quickly with the developed world, resulting in reductions in air pollutant emissions. SSP2 is a so-called "middle of the road scenario" with moderate population growth and slower convergence of income levels across

countries. In SSP2, food consumption especially for resource-intensive livestock based commodities, is expected to increase and energy generation continues to rely on fossil fuels at approximately the same rates as today, resulting in continued growth of GHG emissions. Efforts at curbing air pollution continue along current trajectories with developing economies ultimately catching up to high-income nations, resulting in an eventual decrease in pollutant emissions. Finally, SSPs 3 and 4 depict futures with high inequality between countries (i.e., "regional rivalry") and within countries, respectively. Global GDP growth is low

in both scenarios and concentrated in currently high-income nations whereas population increase is focused in low and middle-income countries. Energy systems in SSP3 see a resurgence of coal dependence whereas reductions occur in SSP4 as the high-tech energy and economy sectors see increased developments and investments leading to higher diversification of technologies (Bauer et al., 2017). Policy making (either regionally or internally) in areas including land-use regulation, air pollution control, and GHG emission limits are less effective. Thus policies vary regionally in both SSPs with weak international institutions

resulting in the highest levels of pollutant and aerosol emissions.

A matrix of socioeconomic-climate scenarios relevant to the broad scientific community was created with SSPs on one axis and climate policy futures (i.e. mitigation scenarios) delineated by end-of-century (EOC) RF on the other axis (see Figure 1). The scenarios selected for inclusion in ScenarioMIP, shown in Table 1, are comprised of both baseline and mitigation cases, in which long-term climate policies are lacking or included, respectively. They are divided into Tier-1 scenarios, which span a

wide range of uncertainty in future forcing and are utilized by other MIPs, and Tier-2 scenarios, which enable more detailed studies of the effect of mitigation and adaptation policies which fall between the Tier-1 forcing levels. Each scenario is run by a single model within ScenarioMIP, comprised of the AIM/CGE, GCAM4, IMAGE, MESSAGE-GLOBIOM, and REMIND-MAgPIE modeling teams. We provide a short discussion here on their selection and refer the reader to (O'Neill et al., 2016, Section 3.2.2) for fuller discussion of the experimental design.

The Tier-1 scenarios include SSP1-2.6, SSP2-4.5, SSP3-7.0, and SSP5-8.5, designed to provide a full range of forcing targets similar in both magnitude and distribution to the RCPs as used in CMIP5. Each EOC forcing level is paired with a specific SSP which is chosen based on the relevant experimental coverage. For example, SSP2 is chosen for the $4.5\,\mathrm{W\,m^{-2}}$ experiment because of its high relevance as a reference scenario to IAV communities as a scenario with intermediate vulnerability and climate forcing and its median positioning of land use and aerosol emissions (of high importance for DAMIP and DCPP)

whereas SSP3 is chosen for the $7.0\,\mathrm{W\,m^{-2}}$ experiment as it allows for quantification of avoided impacts (e.g. relative to SSP2)



**Table 1.** All scenarios and associated attributes used in the ScenarioMIP experiment ensemble.

| Scenario Name | SSP | Target Forcing Level (W m$^{-2}$) | Scenario Type | Tier | IAM | Contributing to other MIPs |
|---|---|---|---|---|---|---|
| SSP1-1.9 | 1 | 1.9 | Mitigation | 2 | IMAGE | ScenarioMIP |
| SSP1-2.6 | 1 | 2.6 | Mitigation | 1 | IMAGE | ScenarioMIP |
| SSP2-4.5 | 2 | 4.5 | Mitigation | 1 | MESSAGE-GLOBIOM | ScenarioMIP, VIACS AB, CORDEX, GeoMIP, DAMIP, DCPP |
| SSP3-7.0 | 3 | 7 | Baseline | 1 | AIM/CGE | ScenarioMIP, AerChemMIP, LUMIP |
| SSP3-LowNTCF | 3 | 6.3 | Mitigation | 2 | AIM/CGE | ScenarioMIP, AerChemMIP, LUMIP |
| SSP4-3.4 | 4 | 3.4 | Mitigation | 2 | GCAM4 | ScenarioMIP |
| SSP4-6.0 | 4 | 6 | Mitigation | 2 | GCAM4 | ScenarioMIP, GeoMIP |
| SSP5-3.4-OS | 5 | 3.4 | Mitigation | 2 | REMIND-MAGPIE | ScenarioMIP |
| SSP5-8.5 | 5 | 8.5 | Baseline | 1 | REMIND-MAGPIE | ScenarioMIP, C4MIP, GeoMIP, ISMIP6, RFMIP |

and has significant emissions from near-term climate forcing (NTCF) species such as aerosols and methane (also referred to as Short-Lived Climate Forcers, or SLCF).

The Tier-2 scenarios include SSP1-1.9, SSP3-LowNTCF, SSP4-3.4, SSP4-6.0, and SSP5-3.4-Overshoot (OS), chosen to both complement and extend the types of scenarios available to climate modelers beyond those analyzed in CMIP5. SSP1-1.9 provides the lowest estimate of future forcing matching the most ambitious goals of the Paris Agreement (i.e., "pursuing efforts to limit the [global average] temperature increase to 1.5°C above pre-industrial levels"). The SSP3-LowNTCF scenario provides an important experimental comparison to scenarios with high NTCFs for use in AerChemMIP (Collins et al., 2017) contrasting with SSP3-7.0. Both SSP4 scenarios fill gaps in Tier-1 forcing pathways and allow investigations of impacts in scenarios with relatively strong land use and aerosol climate effects but relatively low challenges to mitigation. Finally, SSP5-3.4-OS allows for the study of a scenario in which there is large overshoot in RF by mid-century followed by the implementation of substantive policy tools to limit warming in the latter half of the century. It is specifically designed to be twinned with SSP5-8.5, following the same pathway through 2040, and support experiments examining delayed climate action.



### 2.1.1 Historical Emissions Data

We construct a common dataset of historical emissions for the year 2015[1], the transition year in CMIP6 between historic and future model runs, using two primary sources developed for CMIP6. Hoesly et al. (2018) provides data over 1750-2014 for anthropogenic emissions by country. They include a detailed sectoral representation (59 sectors in total) which has been aggre-

gated into nine individual sectors (see SI Table B1), including Agriculture, Aircraft, Energy, Industry, International Shipping, Residential and Commercial, Solvent Production and Application, Transportation, and Waste. Values for 2015 were approximated by extending fossil fuel consumption using aggregate energy statistics (BP, 2016) and trends in emission factors from the GAINS ECLIPSE V5a inventory (Klimont et al., 2017; Stohl et al., 2015). Sulfur ($SO_x$) emissions in China were trended from 2010 using values from Zheng et al. (2018).

van Marle et al. (2017) provide data on historical emissions from open burning, specifically including burning of Agricultural Waste on Fields (AWB), Forests, Grasslands, and Peatlands out to 2015. Due to the high amount of inter-annual variability in the historical data which is not explicitly modeled in IAMs, we use a decadal mean over 2005-2014 to construct a representative value for 2015. When used in conjunction with model results, we aggregate country-level emissions to the individual model regions of which they are comprised.

Emissions of $N_2O$ and fluorinated gas species were harmonized only at the global level, with 2015 values from other data sources. Global $N_2O$ emissions were taken from PRIMAP (Gütschow et al., 2016) and global emissions of HFCs were developed by Velders et al. (2015). The HFC-23 and total PFC and SF6 emissions were provided by Guus Velders, based on Carpenter et al. (2014) mixing ratios and were extended from 2012 to 2015 by using the average 2008-2012 trend.

### 2.1.2 Automated Emissions Harmonization

Emissions harmonization is defined as a procedure designed to match model results to a common set of historical emissions trajectories. The goal of this process is to match a specified base-year dataset while retaining consistency with the original model results to the best extent possible while also providing a smooth transition from historical trajectories. This non-disjoint transition is critical for global climate models when modeling projections of climate futures which depend on historical model runs, guaranteeing a smooth functional shape of both emissions and concentration fields between the historical and future runs.

Models differ in their 2015 data points in part because the historical emissions datasets used to calibrate the models differ (e.g., PRIMAP (Gütschow et al., 2016), EDGAR (Crippa et al., 2016), CEDS (Hoesly et al., 2018)). Another cause of differences is that 2015 is a projection year for all of these models (the original scenarios were originally finalized in 2015).

Harmonization can be simple in cases where a model's historical data is similar to the harmonization dataset. However, when there are strong discrepancies between the two datasets, the choice of harmonization method is crucial for balancing the dual

goals of accurate representation of model results and reasonable transitions from historical data to harmonized trajectories.

The quantity of trajectories requiring harmonization increases the complexity of the exercise. In this analysis, given the available sectoral representation of both the historical data and models, we harmonize model results for 14 individual emissions

---

[1]For sulfur emissions in China, we include values up to 2017, due to a drastic reduction in these emissions in the most recently available datasets.



**Table 2.** Harmonized Species and Sectors, adapted from Gidden et al. (2018) with permission of the authors. A mapping of original model variables (i.e., outputs) to ScenarioMIP sectors is shown in SI Table B2.

| Emissions Species | Sectors |
|---|---|
| Black Carbon (BC) | Agricultural Waste Burning[c] |
| Hexafluoroethane ($C_2F_6$) [a] | Agriculture[c] |
| Tetrafluoromethane ($CF_4$) [a] | Aircraft [b] |
| Methane ($CH_4$) | Energy Sector |
| Carbon Dioxide ($CO_2$) [c] | Forest Burning[c] |
| Carbon Monoxide (CO) | Grassland Burning[c] |
| Hydrofluorocarbons (HFCs) [a] | Industrial Sector |
| Nitrous Oxide ($N_2O$) [a] | International Shipping[b] |
| Ammonia ($NH_3$) | Peat Burning[c] |
| Nitrogen Oxides ($NO_x$) | Residential Commercial Other |
| Organic Carbon (OC) | Solvents Production and Application |
| Sulfur Hexafluoride ($SF_6$) [a] | Transportation Sector |
| Sulfur Oxides ($SO_x$) | Waste |
| Volatile Organic Compounds (VOCs) | |

[a] Global total trajectories are harmonized due to lack of detailed historical data.

[b] Global sectoral trajectories are harmonized due to lack of detailed historical data.

[c] A global trajectory for AFOLU $CO_2$ is used; non-land-use sectors are harmonized for each model region.

**Table 3.** The number of model regions and total harmonized emissions trajectories for each IAM participating in the study. The number of trajectories are calculated from Table 2, including gas species for which global trajectories are harmonized.

| Model | Regions | Harmonized Trajectories |
|---|---|---|
| AIM/CGE | 17 | 1486 |
| GCAM4 | 32 | 2776 |
| IMAGE | 26 | 2260 |
| MESSAGE-GLOBIOM | 11 | 970 |
| REMIND-MAGPIE | 11 | 970 |

species and 13 sectors as described in Table 2. The majority of emissions-sector combinations are harmonized for every native model region (Table 3). Global trajectories are harmonized for fluorinated species and $N_2O$, aircraft and international shipping sectors, and $CO_2$ agriculture, forestry, and other land-use (AFOLU) emissions due to historical data availability and regional detail. Therefore between 970 and 2776 emissions trajectories require harmonization for any given scenario depending on the model used.



We employ the newly available open-source software *aneris* (Gidden et al., 2018; Gidden, 2017) in order to perform harmonization in a consistent and rigorous manner. For each trajectory to be harmonized, *aneris* chooses which harmonization method to use by analyzing both the relative difference between model results and harmonization historical data as well as the behavior of the modeled emissions trajectory. Available methods include ratio and offset methods, which utilize the quotient

and difference of unharmonized and harmonized values respectively, as well as convergence methods which converge to the original modeled results at some future time period. We refer the reader to Gidden et al. (2018) for a full description of the harmonization methodology and implementation.

Override methods can be specified for any combination of species, sectors, and regions which are used in place of the default methods provided by *aneris*. Override methods are useful when default methods do not fully capture either the regional

or sectoral context of a given trajectory. Most commonly, we observed this in cases where there are large relative differences in the historical datasets, the base-year values are small, and there is substantial growth in the trajectory over the modeled time period, thereby reflecting the large relative difference in the harmonized emissions results. However, the number of required override methods is small: 5.1% of trajectories use override messages for the IMAGE model, 5.6% for MESSAGE-GLOBIOM, and 9.8% for REMIND. The AIM model elected not to use override methods, and GCAM uses a relatively large number (35%).

Finally, in order to provide additional detail for fluorinated gases (F-gases) we extend the set of reported HFCs and CFCs species based on exogenous scenarios. We take scenarios of future HFCs from Velders et al. (2015) which provide detailed emissions trajectories for F-gases. We downscale the global HFC emissions reported in each harmonized scenario to arrive at harmonized emissions trajectories for all constituent F-gases, deriving the HFC-23 from the RCP emission pathway. We further include trajectories of CFCs as reported in scenarios developed by the World Meteorological Organization (WMO) (Carpenter

et al., 2014) which are not included in all model results.

### 2.1.3 Region-to-Country Downscaling

Downscaling, defined here as distributing aggregated regional values to individual countries, is performed for all scenarios in order to improve the spatial resolution of emissions trajectories, and as a prelude to mapping to a spatial grid (discussed in SI Section C). We developed an automated downscaling routine that differentiates between two classes of sectoral emissions:

those related to AFOLU and those related to fuel combustion and industrial and urban processes. In order to preserve as much of the original model detail as possible, the downscaling procedures here begin with harmonized emission data at the level of native model regions and the aggregate sectors (Table 2). Here we discuss key aspects of the downscaling methodology and refer the reader to the downscaling documentation[2] for further details.

AFOLU emissions, including Agricultural Waste Burning, Agriculture, Forest Burning, Peat Burning, and Grassland Burn-

ing are downscaled using a linear method. Linear downscaling means that the fraction of regional emissions in each country stays constant over time. (Note that Peat Burning emissions were not modeled by the IAMs and are constant into the future.)

All other emissions are downscaled using the Impact, Population, Affluence, and Technology (IPAT) (Ehrlich and Holdren, 1971) based method developed by van Vuuren et al. (2007), where population and GDP trajectories are taken from the SSP

---

[2]https://github.com/iiasa/emissions_downscaling/wiki





scenario specifications (KC and Lutz, 2014; Dellink et al., 2015). The overall philosophy behind this method is to assume that emission intensity values (i.e., the ratio of emissions to GDP) for countries within a region will converge from a base year, $t_i$ (2015 in this study), over the future. A convergence year, $t_f$, is specified beyond 2100, the last year for the downscaled data, meaning that emission intensities do not converge fully by 2100. The choice of convergence year reflects the rate at which

economic and energy systems converge toward similar structures within each native model region. Accordingly, the SSP1 and SSP5 scenarios are assigned relatively near-term convergence years of 2125, while SSP3 and SSP4 scenarios are assigned 2200, and SSP2 an intermediate value of 2150.

The downscaling method first calculates an emission intensity, $I$, for the base and convergence years using emission level, $E$, and $GDP$.

$$I_t = \frac{E_t}{GDP_t} \qquad (1)$$

An emission intensity growth rate, $\dot{I}$, is then determined for each country, $c$, within a model region, $R$, using convergence year emission intensities, $I_{R,t_f}$, determined by extrapolating from growth rates over the last 10 years (e.g., 2090 to 2100) of the scenario data.

$$\dot{I}_c = \frac{I_{R,t_f}}{I_{c,t_i}}^{\frac{1}{t_f - t_i}} \qquad (2)$$

Using base-year data for each country and scenario data for each region, future downscaled emission intensities and patterns of emissions are then generated for each subsequent time period.

$$I_{c,t} = \dot{I}_c I_{c,t-1} \qquad (3)$$

$$E^*_{c,t} = I_{c,t} GDP_{c,t} \qquad (4)$$

These spatial patterns are then scaled with the model region data to guarantee consistency between the spatial resolutions,

resulting in downscaled emissions for each country in each time period

$$E_{c,t} = \frac{E_{R,t}}{\sum_{c' \in R} E^*_{c',t}} E^*_{c,t} \qquad (5)$$

For certain countries and sectors the historical dataset has zero-valued emissions in the harmonization year. This would result in zero downscaled future emissions for all years. Zero emissions data occurs largely for small countries, many of them small island nations. This could either be due to lack of actual activity in the base-year, or missing data on activity in those



countries. In order to allow for future sectoral growth in such cases, we adopt, for purposes of the above calculations, an initial emission intensity of $\frac{1}{3}$ the value of the lowest country in the same model region. We then allocate future emissions in the same manner discussed above, which is consistent with our overall convergence assumptions. Note that we exclude the industrial sector (Table 2) from this operation as it might not be reasonable to assume the development of substantial industrial activity

in these countries.

Finally, some scenarios (notably energy) include negative $CO_2$ emissions at some point in the future. For $CO_2$ emissions, therefore, we apply a linear rather than exponential function to allow a smooth transition to negative emissions values for both the emissions intensity growth rate and future emission intensity calculations. In such cases, Equations 2 and 3 are replaced by 6 and 7, respectively.

$$\dot{I_c} = \left( \frac{I_{R,t_f}}{I_{c,t_i}} - 1 \right) \frac{1}{t_f - t_i} \tag{6}$$

$$I_{c,t} = (1 + \dot{I_c})I_{c,t-1} \tag{7}$$

## 3    Results

Here we present the results of harmonization and downscaling applied to all nine scenarios under consideration. We discuss in Section 3.1 the relevance of each selected scenario to the overall experimental design of ScenarioMIP, focusing on their RF

and mean global temperature pathways. In Section 3.2, we discuss general trends in global trajectories of important GHGs and aerosols and their sectoral contributions over the modeled time horizon. In Section 3.3, we explore the effect of harmonization on model results and the difference between unharmonized and harmonized results. Finally, in Section 3.4, we provide an overview of the spatial distribution of emissions species at both regional and spatial grids.

### 3.1    Experimental Design and Global Climate Response

The nine ScenarioMIP scenarios were selected to provide a robust experimental design space for future climate studies as well as IAV analyses with the broader context of CMIP6. Chief among the concerns in developing such a design space is both the range and spacing of the global climate response within the portfolio of scenarios(Moss et al., 2008). Prior work for the RCPs studied a range of climate outcomes between ~2.6-8.5 $\mathrm{W\,m^{-2}}$ at EOC. Furthermore, recent work (Tebaldi et al., 2015) finds that statistically significant regional temperature outcomes (>5% of half the land surface area) are observable with a minimum

separation of 0.3°C, which is approximately equivalent to 0.75$\mathrm{W\,m^{-2}}$ (O'Neill et al., 2016). Given the current policy context, notably the recent adoption of the UN Paris Agreement, the primary design goal for the ScenarioMIP scenario selection is thus twofold: span a wider range of possible climate futures (1.9-8.5 $\mathrm{W\,m^{-2}}$) in order to increase relevance to the global climate dialogue and provide a variety of scenarios between these upper and lower bounds such that they represent statistically significant climate variations in order to support a wide variety of CMIP6 analyses.





We find that the selected scenarios meet this broad goal, as shown in Figure 2, by using the simple climate model MAGICC6 with central climate-system and gas-cycle parameter settings for all scenarios to calculate pathways of both RF and the resulting response of global mean temperature (see SI Table B3 for a listing of all EOC RF values).

We also present illustrative global-mean temperature pathways. EOC temperature outcomes span a large range, from 1.4°C
at the lower end to 4.9°C for SSP5-8.5, the scenario with highest warming emissions trajectories. Notably, two scenarios (SSP1-1.9, which reaches 1.4°C by EOC and SSP1-2.6, reaching 1.7°C) can be used for studies of global outcomes of the implementation of the UN Paris Agreement, which has a desired goal of "[h]olding the increase in the global average temperature to well below 2°C above pre-industrial levels and pursuing efforts to limit the temperature increase to 1.5°C above pre-industrial levels" (United Nations, 2016, Article 2.1(a)). The difference between scenario temperature outcomes is statistically significant
in nearly all cases, with a minimum difference of 0.37°C (SSP1-1.9 and SSP1-2.6) and maximum value of 0.77°C (SSP3-7.0 and SSP5-8.5). The EOC difference between SSP4-3.4 and SSP5-3.4-OS is not significant (0.07°C); however global climate outcomes are likely sensitive to the dynamics of the forcing pathway (Tebaldi et al., 2015).

A subset of four scenarios (SSP1-2.6, SSP2-4.5, SSP4-6.0, and SSP5-8.5) were also designed to provide continuity between CMIP5 and CMIP6 by providing similar forcing pathways to their RCP counterparts assessed in CMIP5. We find that this
aspect of the scenario design space is also met by the relevant scenarios. SSP2-4.5 and SSP5-8.5 track RCP4.5 and RCP8.5 pathways nearly exactly. We observe slight deviations between SSP1-2.6 and RCP2.6 as well as SSP4-6.0 and RCP6.0 at mid-century due largely to increased methane emissions in the historic period (i.e., methane emissions broadly follow RCP8.5 trajectories after 2000 resulting in higher emissions in the harmonization year of this exercise; see Figure 3 below).

The remaining five scenarios were chosen to "fill gaps" in the previous RCP studies in CMIP5 and enhance the potential
policy relevance of CMIP6 MIP outputs (O'Neill et al., 2016). SSP3-7.0 was chosen to provide a scenario with relatively high vulnerability and land use change with associated near-term climate forcer (NTCF) emissions resulting in a high RF pathway. We find that it reaches an EOC forcing target of ~7.1 $\mathrm{W\,m^{-2}}$ and greater than 4°C mean global temperature increase. While contributions to RF from $CO_2$ in SSP3-7.0 are lower than that of SSP5-8.5, methane and aerosol contributions are considerably higher. A companion scenario, SSP3-LowNTCF, was also included in order to study the effect of NTCF species in the context
of AerChemMIP. Critically, emissions factors of key NTCF species are assumed to develop similar to an SSP1 (rather than SSP3) scenario. SSP3-LowNTCF sees substantially less contributions to EOC forcing from NTCF emissions (notably $SO_x$ and methane), resulting in a forcing level of 6.3 $\mathrm{W\,m^{-2}}$ and global mean temperature increase of 3.75°C by the end of the century. This significant reduction is largely due to updating emissions coefficients for air pollutants and other NTCF to match the SSP1 assumptions. SSP4-3.4 was chosen to provide a scenario at the lower end of the range of future forcing pathways.
Reaching a EOC mean global temperature between SSP2-4.5 and SSP1-2.6 (~2.25°C), it is an ideal scenario for scientists to study the mitigation costs and associated impacts between forcing levels of 4.5 and 2.6 $\mathrm{W\,m^{-2}}$.

The final two scenarios, SSP1-1.9 and SSP5-3.4-OS were chosen to study policy-relevant questions of near and medium-term action on climate change. SSP1-1.9 provides a new low-end to the RF pathway range. It reaches an EOC forcing level of ~1.9 $\mathrm{W\,m^{-2}}$ and an associated global mean temperature increase of ~1.4°C (with temperature peaking in 2040), in line with
the goals of the Paris Agreement. SSP5-3.4-OS, on the other hand, is designed to represent a world in which action towards





**Figure 2.** Trajectories of RF and global mean temperature (above pre-industrial levels) are presented as are the contributions to RF for a number of different emissions types native to the MAGICC6 model. The RF trajectories are displayed with their RCP counterparts analyzed in CMIP5. For those scenarios with direct analogues, trajectories are largely similar in shape and match the same EOC forcing values.




climate change mitigation is delayed but vigorously pursued after 2050, resulting in a forcing and mean global temperature "overshoot". A peak temperature of 2.5°C above pre-industrial levels is reached in 2060 after which global mitigation efforts reduce EOC warming to ~2.25°C. In tandem, and including SSP2-4.5 (which serves as a reference experiment in ScenarioMIP (O'Neill et al., 2016)), these scenarios provide a robust experimental platform to study the effect of the timing and magnitude

of global mitigation efforts which can be especially relevant to science-informed policy discussions.

### 3.2 Global Emissions Trajectories

Emissions contributions to the global climate system are myriad but can broadly be divided into contributions from Greenhouse Gases (GHGs) and aerosols. The models used in this analysis explicitly represent manifold drivers and processes involved in the emissions of various gas species. For a fuller description of these scenario results see the original SSP quantification papers

(van Vuuren et al., 2017; Fricko et al., 2017; Fujimori et al., 2017; Calvin et al., 2017; Kriegler et al., 2017). Here, we focus on emissions species that most strongly contribute to changes in future mean global temperature and scenarios with the highest relevance and uptake for other MIPs within CMIP6, namely the Tier-1 scenarios SSP1-2.6, SSP2-4.5, SSP3-7.0, and SSP5-8.5. Where insightful, we provide additional detail on results from other scenarios; however results for all scenarios are available in SI Section D.

$CO_2$ emissions have a large span across scenarios by the end of the century (-20 Gt/yr to 125 Gt/yr), as shown in Figure 3. Scenarios can be categorized based on characteristics of their trajectory profiles: those that have consistent downwards trajectories (SSP1, SSP4-3.4), those that peak in a given year and then reduce in magnitude (SSP2-4.5 in 2040 and SSP4-6.0 in 2050), and those that have consistent growth in emissions (SSP3). SSP5 scenarios, which model a world with fossil-fuel driven development, have EOC emissions which bound the entire scenario set, with the highest $CO_2$ emissions in SSP5-8.5

peaking in 2080 and the lowest $CO_2$ emissions in SSP5-3.4-OS resulting from the application of stringent mitigation policies after 2040 in an attempt to stabilize RF to 3.4 $\mathrm{W\,m^{-2}}$ after overshooting this limit earlier in the century. A number of scenarios exhibit negative net $CO_2$ emissions before the end of the century. SSP1-1.9, the scenario with the most consistent negative emission trajectory, first reports net negative emissions in 2060 with EOC emissions of -14 Gt/yr. SSP5-3.4-OS, SSP1-2.6, and SSP4-3.4 each cross the 0-emissions threshold in 2070, 2080, and 2090, respectively.

Global emissions trajectories for $CO_2$ are driven largely by the behavior of the energy sector in each scenario, as shown in Figure 4. Positive emissions profiles are also greatly influenced by the industry and transport sectors whereas negative emissions profiles are driven by patterns of agriculture and land-use as well as the means of energy production. In SSP1-2.6, early-mid century emissions continue to be dominated by the energy sector with substantial contributions from industry and transport. Negative emissions from land use are observed as early as 2030 due to large-scale afforestation (Popp et al., 2017; van Vuuren

et al., 2017) while net negative emissions from energy conversion first occur in 2070. Such net negative emissions are achieved when carbon dioxide removal from bioenergy with CCS exceeds residual fossil $CO_2$ emissions from the combustion of coal, oil and gas. Emissions contributions from the transport sector diminish over the century as heavy and light-duty transport fleets are electrified. Emissions from industry peak and then reduce over time such that the residential and commercial sector (RC) provides the majority of positive $CO_2$ emissions by the end of the century. SSP2-4.5 experiences similar trends among sectors





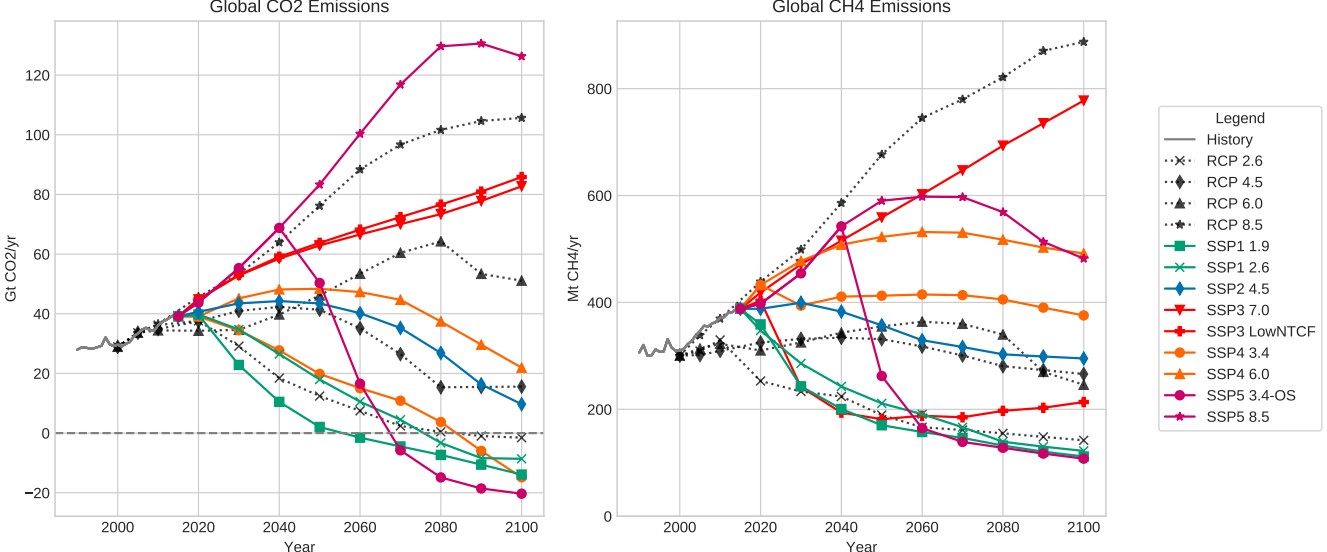

**Figure 3.** Trajectories of $CO_2$ and $CH_4$, primary contributors to GHG emissions, including both historical emissions, emissions analyzed for the RCPs, and all nine scenarios covered in this study.

but with smaller magnitudinal changes and temporal delays. Negative emissions, for example, are experienced in the land-use sector for the first time in 2060 and are not experienced in the energy sector until the end of the century. Energy-sector $CO_2$ emissions continue to play a large role in the overall composition until 2080, at which point the industrial sector provides the plurality of $CO_2$. Emissions from the transport sector peak at mid-century, but are still a substantive component of positive

$CO_2$ emissions at the end of the century. Finally, the SSP5-8.5 scenario's emission profile is dominated by the fossil-fueled energy sector for the entirety of the century. Contributions from the transport and industrial sectors grow in magnitude but are diminished as share of total $CO_2$ emissions, $CO_2$ emissions from the AFOLU sector decrease steadily over time. By the end of the century, the energy sector comprises almost 75% of all emitted $CO_2$ in this scenario relative to 50% today.

    Methane ($CH_4$) is an emissions species with substantial contributions to potential future warming mainly due to its immedi-

ate GHG effect, but also because of its influence on atmospheric chemistry, as a tropospheric ozone precursor, and its eventual oxidation into $CO_2$ in the case of $CH_4$ from fossil sources (Boucher et al., 2009). At present, approximately 400 Mt/yr of $CH_4$ is emitted globally, and the span of future emissions developed in this scenario set range from 100 to nearly 800 Mt/yr by the end of the century. Global emissions of methane in SSP1 scenarios follow similar trajectories to $CO_2$, with large emissions reductions; SSP2 follows suit, with emissions peaking in 2030 and then reducing throughout the rest of the century; in SSP3's

baseline scenario, emissions continue to grow while in the NTCF scenario, they are reduced drastically as policies are implemented to reduce forcing from short-lived emissions species; SSP4 is characterized by growing (SSP4-6.0) or mostly stable (SSP4-3.4) $CH_4$ emissions until the middle of the century which peak in 2060 and then decline; and finally SSP5's baseline



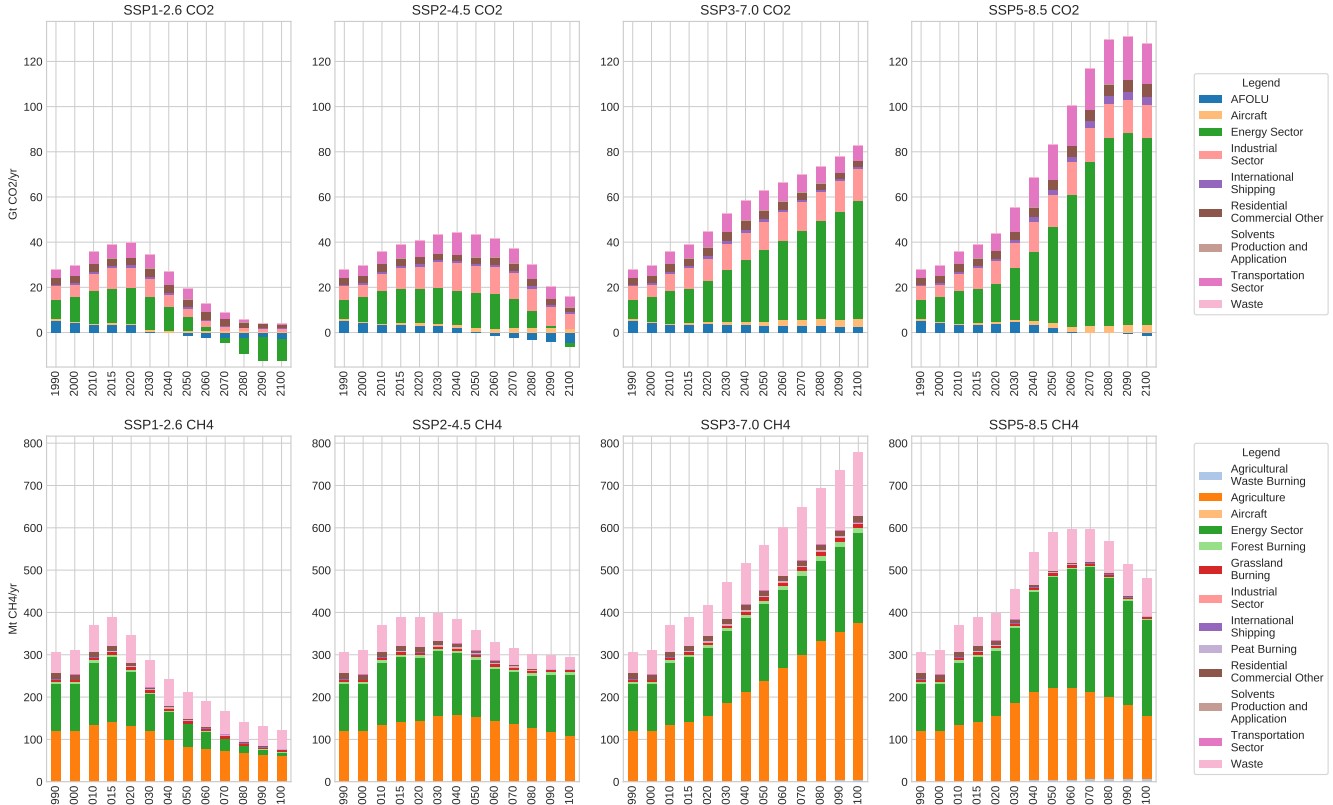

**Figure 4.** The sectoral contributions to $CO_2$ and $CH_4$ emissions for Tier-1 scenarios.

scenario sees a plateauing of $CH_4$ emissions between 2050 and 2070 before their eventual decline while the overshoot scenario has drastic $CH_4$ emissions reductions in 2040 corresponding to significant mid-century mitigation efforts in that scenario.

Historically, $CH_4$ emissions are dominated by three sectors: energy (due to fossil fuel production, and natural gas transmission), agriculture (largely enteric fermentation from livestock and rice production), and waste (i.e., landfills). In each scenario, global emissions of $CH_4$ are largely dominated by the behavior of activity in each of these sectors over time. For example, in the SSP1 scenarios, significant reductions in energy emissions are observed as energy supply systems shift from fossil to renewable sources while agriculture and waste-sector emissions see only modest reductions as global population stabilizes around mid-century. In the SSP2 scenario, emissions from the energy sector peak in 2040 as there is continued reliance on energy from natural gas but large expansions in renewables in the future; however, emissions from the agricultural and waste sectors are similar to today's levels by the end of the century. Finally, $CH_4$ emissions in SSP5's baseline scenario is characterized by growth in energy sector from continued expansion of natural gas and a peak and reduction in agricultural emission resulting in 20% higher emissions at the end of the century relative to the present as population grows in the near term before contracting globally.



GHG emissions are broadly similar between the main scenarios in CMIP5 (RCPs) and CMIP6 (SSPs). Notably, we observe that the SSPs exhibit slightly lower $CO_2$ emissions in the 2.6 $W\,m^{-2}$ scenarios and higher emissions in the 8.5 $W\,m^{-2}$ scenarios due to lower and higher dependence on fossil fuels relative to their RCP predecessors. $CH_4$ emissions are largely similar at EOC for 2.6 and 4.5 $W\,m^{-2}$ scenarios between the RCPs and SSPs, with earlier values differing due to continued

growth in the historical period (RCPs begin in 2000 whereas SSPs begin in 2015). The 8.5 $W\,m^{-2}$ scenario exhibits the largest difference in $CH_4$ emissions between the RCPs and SSPs because of the SSP5 socioeconomic story line depicting a world which largely develops out of poverty in less-developed countries, reducing $CH_4$ emissions from waste and agriculture. This contrasts with a very different story line behind RCP-8.5 (Riahi et al., 2011).

In nearly all scenarios, aerosol emissions are observed to decline over the century; however, the magnitude and speed of

this decline is highly dependent on the evolution of various drivers based on the underlying SSP story lines, resulting in a wide range of aerosol emissions, as shown in Figure 5. For example, sulfur emissions (totaling 112 Mt/yr globally in 2015) are dominated at present by the energy and industrial sectors. In SSP1, where the world transitions away from fossil-fuel related energy production (namely coal in the case of sulfur), emissions decline sharply as the energy sector transitions to non-fossil based fuels and end-of-pipe measures for air pollution control are ramped up swiftly. The residual amount of sulfur remaining

at the end of the century (~10 Mt/yr) is dominated by the industrial sector. SSP2-4.5 sees a similar transition but with delayed action: total sulfur emissions decline due primarily to the decarbonization of the energy sector. SSP5 also observes declines in overall sulfur emissions led largely by an energy mix that transitions from coal dependence to dependence on natural gas, as well as strong end-of-pipe air pollution control efforts. These reductions are similarly matched in the industrial sector, where natural gas is substituted for coal use as well. Thus, overall reductions in emissions are realized across the scenario set.

Only SSP3 shows EOC sulfur emissions equivalent to the present day, largely due to increased demand for industrial services from growing population centers in developing nations with a heavy reliance on coal-based energy production, and weak air pollution control efforts.

Aerosols associated with the burning of traditional biomass, crop, and pasture residues, as well as municipal waste, such as black carbon (BC) and organic carbon (OC, see SI Figure D3) are affected most strongly by the degree of economic progress

and growth in each scenario, as shown in Figure 6. For example, BC emissions from the residential and commercial sector comprise nearly 40% of all emissions in the historical time period with a significant contribution from mobile sources. By the end of the century, however, emissions associated with crop and pasture activity comprise the plurality of total emissions in each of SSPs 1, 2, and 5 due to a transition away from traditional biomass usage based on increased economic development and population stabilization and emissions controls on mobile sources. Only SSP3, in which there is continued global inequality

and the persistence of poor and vulnerable urban and rural populations, is there continued quantities of BC emissions across sectors similar to today. OC emissions are largely from biofuel and open burning and follow similar trends: large reductions in scenarios with higher income growth rates with a residual emissions profile due largely to open burning-related emissions. Other pollutant emissions (e.g., $NO_x$, carbon monoxide (CO), and volatile organic carbon (VOC)) also see a decline in total global emissions at rates depending on the story line (Rao et al., 2017).



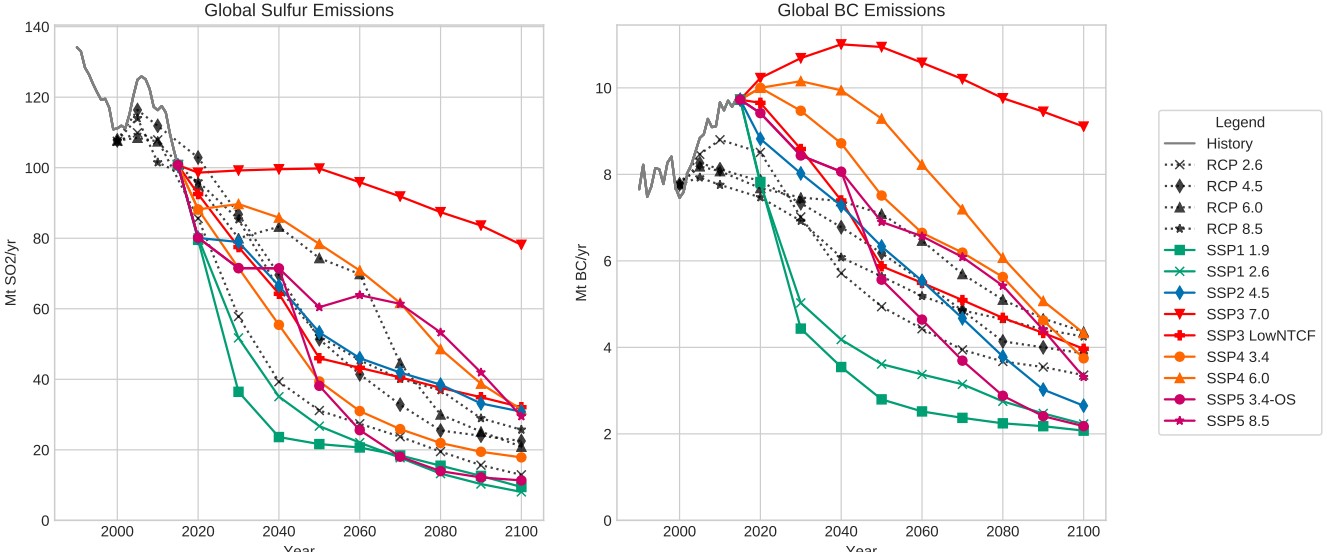

**Figure 5.** Emissions trajectories for sulfur and black carbon (BC), for history, the RCPs, and all nine scenarios analyzed in this study. SSP trajectories largely track with RCP values studied in CMIP5. A notable difference lies in BC emissions, which have seen relatively large increases in past years, thus providing higher initial emissions for the SSPs.

### 3.3 The Effects of Harmonization

Harmonization, by definition, modifies the original model results such that base-year values correspond to an agreed-upon historical source, with an aim for future values to match the original model behavior as much as possible. Model results are harmonized separately for each individual combination of model region, sector, and emissions species. In the majority of cases, model results are harmonized using the default methods described in Section 2.1.2; however, it is possible for models to provide harmonization overrides in order to explicitly set a harmonization method for a given trajectory.

We assess the impact that harmonization has on model results by analyzing the harmonized and unharmonized trajectories. Figure 7 shows global trajectories for each scenario of a selected number of emissions species. Qualitatively, the $CO_2$ and sulfur emissions trajectories match relatively closely to the magnitude of model results due to general agreement between historical sources used by individual models and the updated historical emissions datasets. This leads to convergence harmonization routines being used by default. In the case of $CH_4$ and BC, however, there is larger disagreement between model results and harmonized results in the base year. In such cases, *aneris* chooses harmonization methods that match the shape of a given trajectory rather than its magnitude in order to preserve the relationship between driver and emission for each model.

We find that across all harmonized trajectories the difference between harmonized and unharmonized model results decreases over the modeled time horizon. The lower panel in Figure 7 shows the distribution of all 15,954 trajectories (unharmonized less harmonized result) for the harmonization year (2015) and two modeled years (2050 and 2100). Each emissions species data



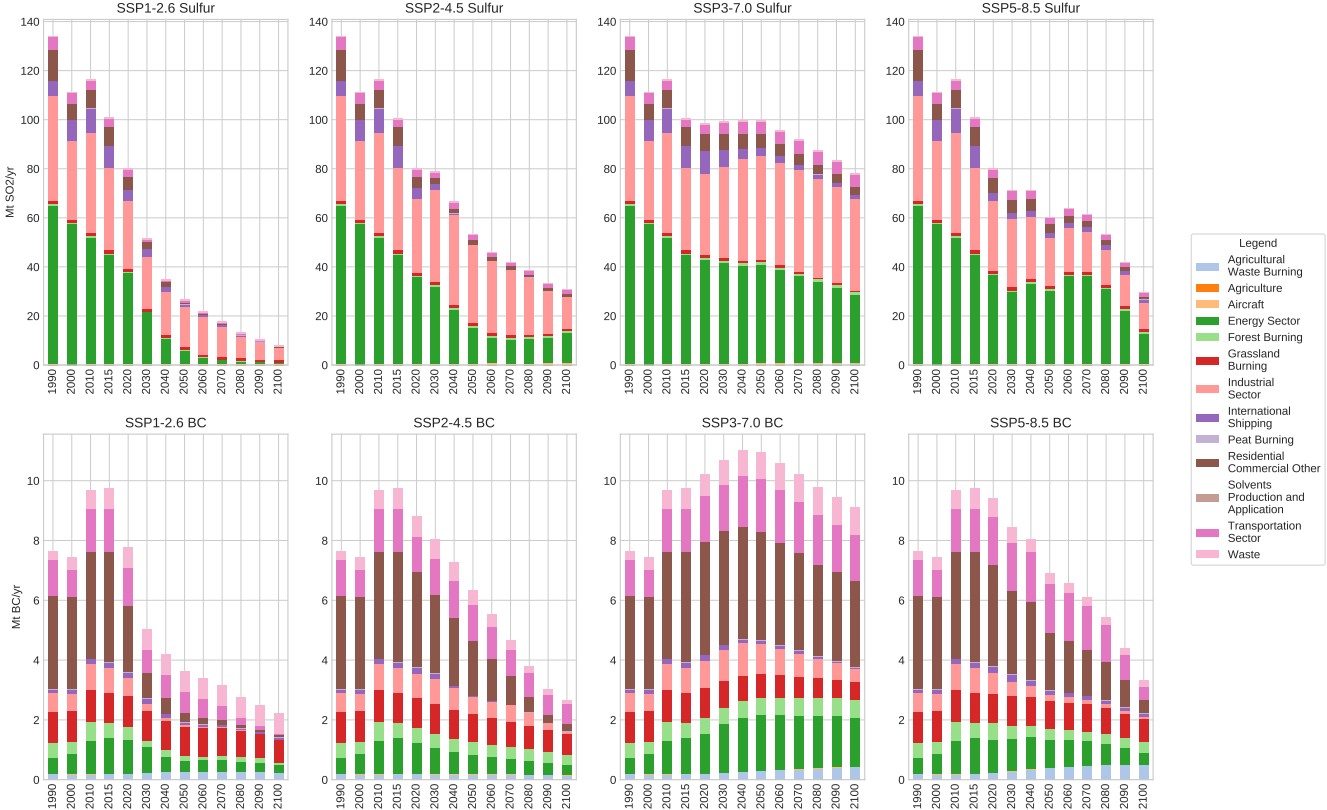

**Figure 6.** The sectoral contributions to sulfur and black carbon emissions for Tier-1 scenarios.

population exhibits the same trend of reduced difference between modeled and harmonized results. Not only do the deviation of result distributions reduce over time, but the median value also converges toward zero in all cases.

The trajectory behavior for a number of important emissions species are dominated by certain sectors, as discussion previously in Section 3.1 and shown in SI Figure E1. Notably, the energy sector tends to dominate behavior of $CO_2$ emissions, agriculture dominates $CH_4$ emission trajectories, the industrial sector largely determines total sulfur emissions, and emissions from the residential and commercial sectors tend to dominate BC emissions across the various scenarios. Accordingly, we further analyzed the harmonization behavior of these sector-species combinations. Importantly, we again observe an overall trend towards convergence of results at the end of the century; thus harmonized results largely track unharmonized results for these critical emissions sectors. The deviation of distributions of differences consistently reduce with time for all scenarios, and nearly all medians converge consistently towards zero, save for energy-related $CO_2$ SSP5-8.5 which has a higher growth rate than convergence rate, thus larger differences in 2050 than 2015. Overall, we find the harmonization procedure successfully harmonized results historical base year and closely matches model results across the scenarios by EOC.





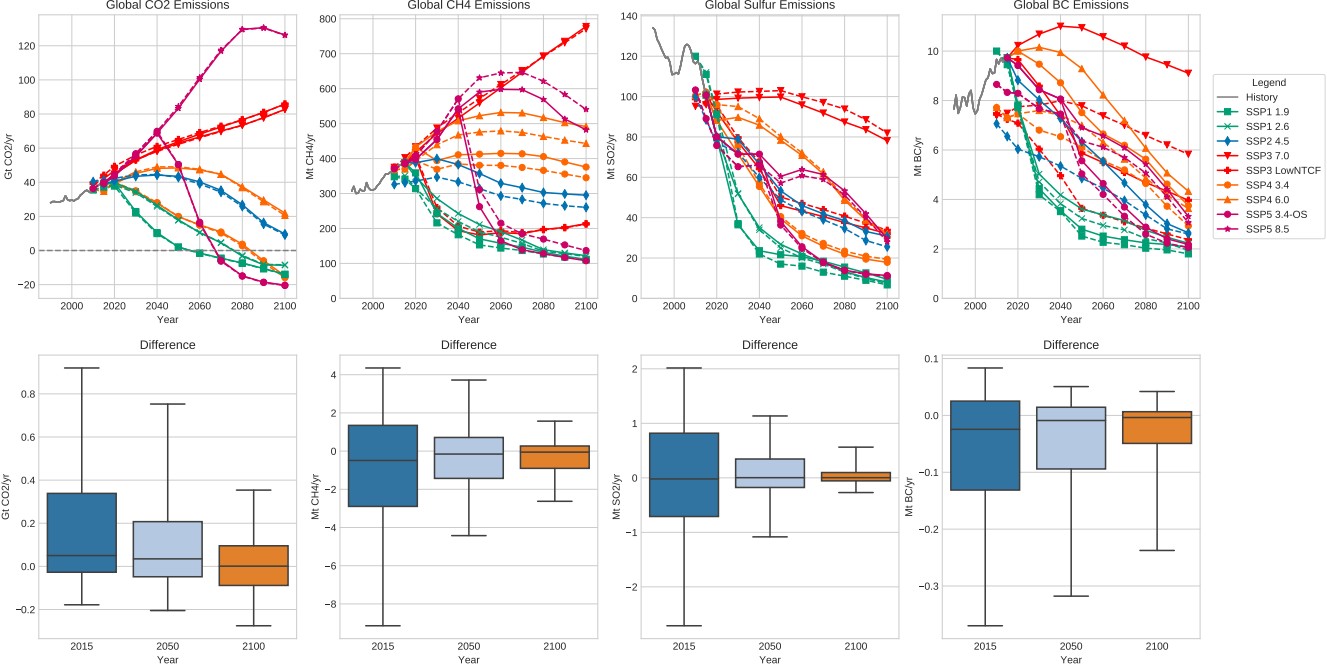

**Figure 7.** Harmonized (solid) and unharmonized (dashed) trajectories are shown in the upper panel. The lower panel depicts the distribution of differences (harmonized less unharmonized) for every modeled region. All box plots show upper and lower quartiles as solid boxes, median values as solid lines, and whiskers extending to 10th and 90th percentiles. Median values for all are near zero, however, the deviation reduces with time as harmonized values begin to more closely match unharmonized model results largely due to the use of convergence methods.

## 3.4 Spatial Distribution of Emissions

The extent to which reductions or growth of emissions are distributed regionally varies greatly among scenarios. The regional breakdown of primary contributors to future warming potential, $CO_2$ and $CH_4$, is shown in Figure 8. While present-day $CO_2$ emissions see near-equal contributions from the OECD and Asia, future $CO_2$ emissions are governed largely by potential

5  developments in Asia (namely China and India). For SSP1-2.6, in which deep decarbonization and negative $CO_2$ emissions occur before the end of the century, emissions in Asia peak in 2020 before reducing to zero by 2080. Mitigation efforts occur across all regions, and the majority of carbon reduction is focused in the OECD; however, all regions have net negative $CO_2$ emissions by 2090. Asian $CO_2$ emissions in SSP2-4.5 peak in 2030, and most other regions see overall reductions except Africa, in which continued development and industrialization results in emissions growth. Notably, Latin America is the only

10  region in which negative emissions occur in SSP2-4.5 due largely to increased deployment of biomass-based energy production and carbon sequestration. Sustained growth across regions is observed in SSP5-8.5, where emissions in Asia peak by 2080,





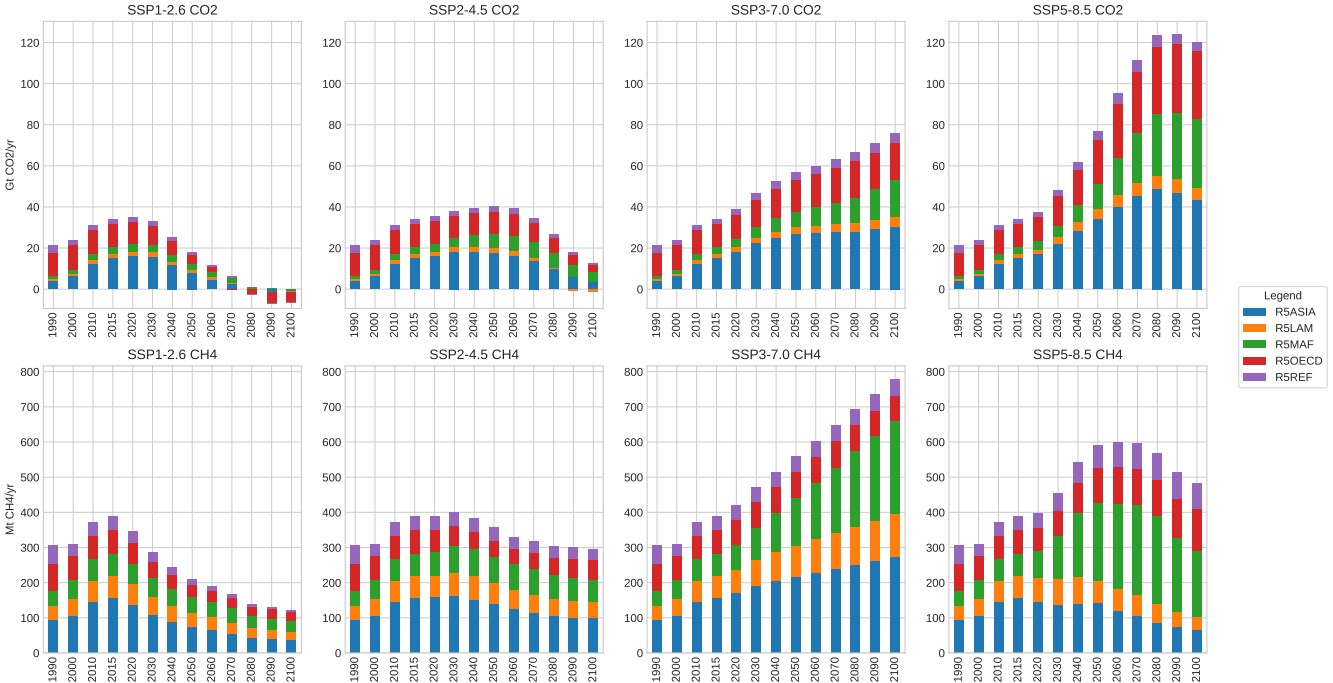

**Figure 8.** Regional emissions for five global regions for $CO_2$ and $CH_4$ in each Tier-1 scenario.

driving the global emissions peaking in the same year. Other scenarios (see SI Figure F1) follow similar trends with future $CO_2$ emissions driven primarily by developments in Asia.

$CH_4$ emissions, resulting from a mix of energy use, food production, and waste disposal, show a different regional break-down across scenarios. In SSP1-2.6, $CH_4$ emissions are reduced consistently across regions as energy systems transition away

from fossil fuel use (notably natural gas) and the husbandry of livestock is curtailed globally. $CH_4$ emissions in other scenarios tend to be dominated by developments in Africa. In SSP5-8.5, for example, emissions in Africa begin to dominate the global profile by mid-century, due largely to expansion of fossil-fuel based energy production. SSPs 3 and 4 see continued growth in African $CH_4$ emissions across the century, even when global emissions are reduced as in the case of SSP4 scenarios.

$CO_2$ and $CH_4$ are well-mixed climate forcers (Stocker et al., 2013) and thus their spatial variation have a higher impact

from a political rather than physical perspective. Aerosols, however, have substantive spatial variability which directly impacts both regional climate forcing via scattering and absorption of solar radiation and cloud formation as well as local and regional air quality. Thus in order to provide climate models with more detailed and meaningful datasets, we downscale emissions trajectories from model regions to individual countries using the methodology described previously in Section 2.4, which are subsequently mapped to spatial grids (Feng, 2018). We here present global maps of two aerosol species with the strongest

implications on future warming, i.e., BC in Figure 9 and sulfur in Figure 10. We highlight three cases which have relevant aerosol emissions profiles: SSP1-2.6 which has significantly decreasing emissions over the century, SSP3-7.0 which has the





highest aerosol emissions, and SSP3-LowNTCF which has similar socioeconomic drivers as the SSP3 baseline but models the inclusion of policies which seek to limit emission of near-term climate forcing species.

At present, BC has the highest emissions in China and India due largely to traditional biomass usage in the residential sector and secondarily to transport-related activity. In scenarios of high socioeconomic development and technological progress, such as SSP1-2.6, emissions across countries decline dramatically such that by the end of the century, total emissions in China, for example, are equal to that of the USA today. In almost all countries, BC emissions are nearly eradicated by mid-century while emissions in southeast Asia reach similar levels by the end of the century. In SSP3-7.0, however, emissions from southeast Asia and central Africa increase until the middle of the century as populations grow while still depending on fossil-heavy energy supply technologies, transportation, and cooking fuels. By the end of the century in SSP3-7.0, global BC emissions are nearly equivalent to the present day (see, e.g., Figure 5), but these emissions are concentrated largely in central Africa, southeast Asia, and Brazil while they are reduced in North America, Europe, and Central Asia. By enacting policies that specifically target near-term climate forcers in SSP3-LowNTCF, the growth of emissions in the developing world is muted by mid-century and are cut by more than half of today's levels (~9 Mt/yr vs. ~4 Mt/yr) by the end of the century. These policies result in similar levels of BC emissions in China as in SSP1-2.6, while most of the additional emissions are driven by activity in India and central Africa due to continued dependence on traditional biomass for cooking and heating.

The spatial distribution of sulfur emissions varies from that of BC due to large contributions from energy and industrial sectors, and thus being driven by a country's economic size and composition, as opposed to household activity. Emissions today are largely concentrated in countries having large manufacturing, industrial, and energy supply sectors with heavy reliance on coal, such as the China, India, the USA, Russia, and some parts of the Middle East. Again, we observe in SSP1-2.6 a near elimination of sulfur emissions by the end of the century with some continued reliance on sulfur-emitting technologies in India and China in the middle of the century. In SSP3-7.0, although global sulfur emissions over the course of the century peak slightly before reducing to below current levels, increased emissions in southeast Asia offset reductions in emissions elsewhere due to an expanding industrial sector with continued reliance on coal. Notably, emissions in India peak around mid-century before reducing to a magnitude lower than emissions levels today. In the SSP3-LowNTCF scenario, NTCF policies have the added effect of reducing sulfur emissions, resulting in more RF but less potential health impacts due to sulfur pollution. By the end of the century in SSP3-LowNTCF, only India, China, and Brazil have non-trivial quantities of emissions at significantly lower magnitudes than today.

## 4 Conclusions

We present a suite of nine scenarios of future emissions trajectories of anthropogenic sources, a key deliverable of the ScenarioMIP experiment within CMIP6. IAM results for 14 different emissions species and 13 individual sectors are provided for each scenario with consistent transitions from the historical data used in CMIP6 to future trajectories using automated harmonization before being downscaled to provide higher emission source spatial detail. Harmonized emissions at global, original





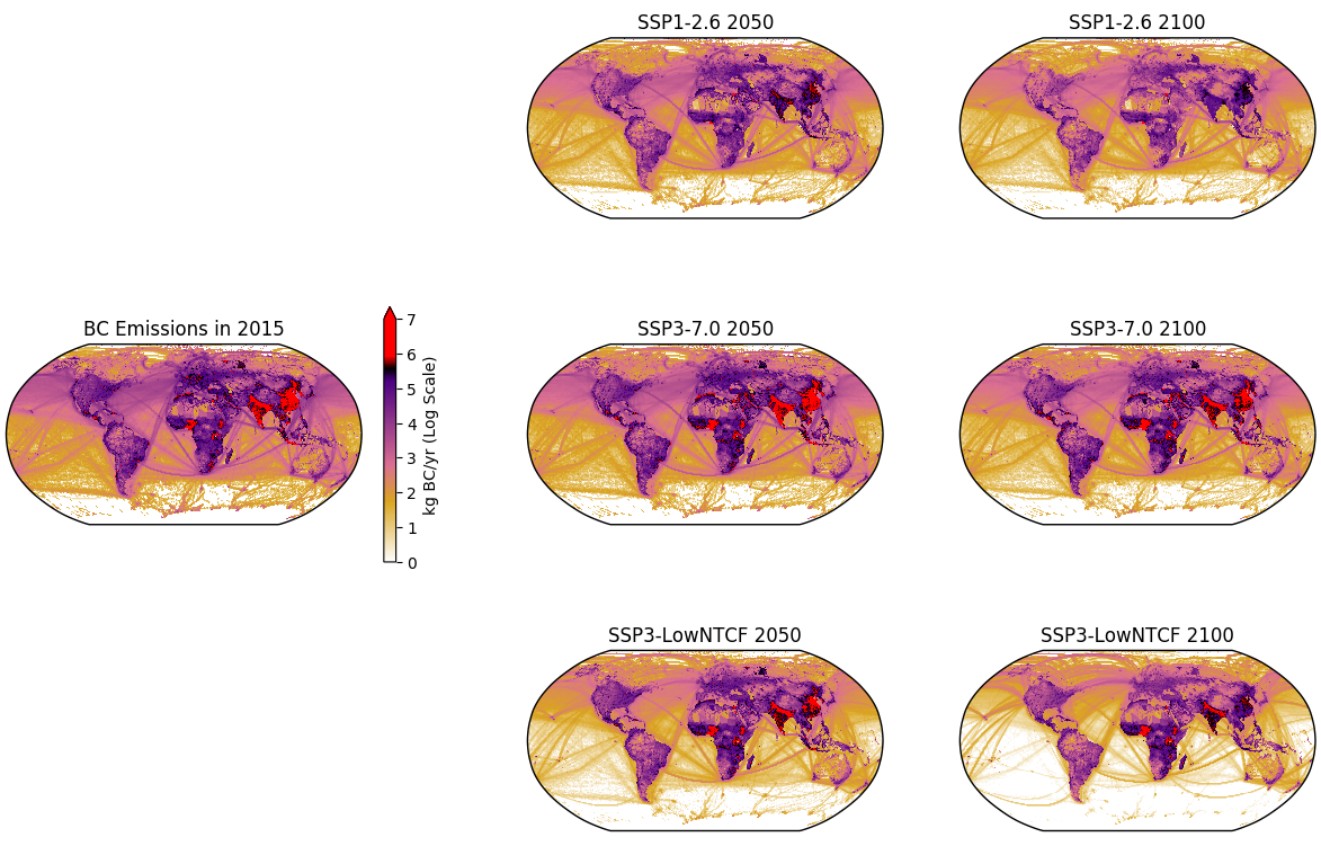

**Figure 9.** Downscaled and gridded emissions of Black Carbon at present and in 2050 and 2100 for SSP1-2.6, SSP3-7.0, and SSP3-LowNTCF.

native model region, and gridded resolution have been delivered to participating climate teams in CMIP6 for further analysis and study by a number of different MIPs.

Scenarios were selected from a candidate pool of over 40 different SSP realizations such that a range of climate outcomes are represented which provide sufficient spacing between EOC forcing to sample statistically significant global and regional temperature outcomes. Of the nine scenarios, four were selected to match forcing levels previously provided by the RCP scenarios used in CMIP5 (RCP2.6 aligns with SSP1-2.6, RCP4.5 with SSP2-4.5, RCP6.0 with SSP4-6.0, and RCP8.5 with SSP5-8.5). RF trajectories are largely comparable between two scenario sets, with relatively strong deviations in the $6.0 \, \mathrm{W \, m^{-2}}$ scenario forcing pathways (SSP4-6.0 has higher emissions in the mid-century period compared with RCP6.0, but both reach similar EOC forcing) and small differences in the $2.6 \, \mathrm{W \, m^{-2}}$ scenarios. Notably, SSP5-8.5 has significantly higher $CO_2$ and lower $CH_4$ emissions compared with RCP8.5, following SSP5's fossil-fuel-based and agricultural intensification SSP5 story line; however, their forcing trajectories are nearly identical. SSP2-4.5 is observed to have a strongly consistent forcing pathway compared to RCP4.5.





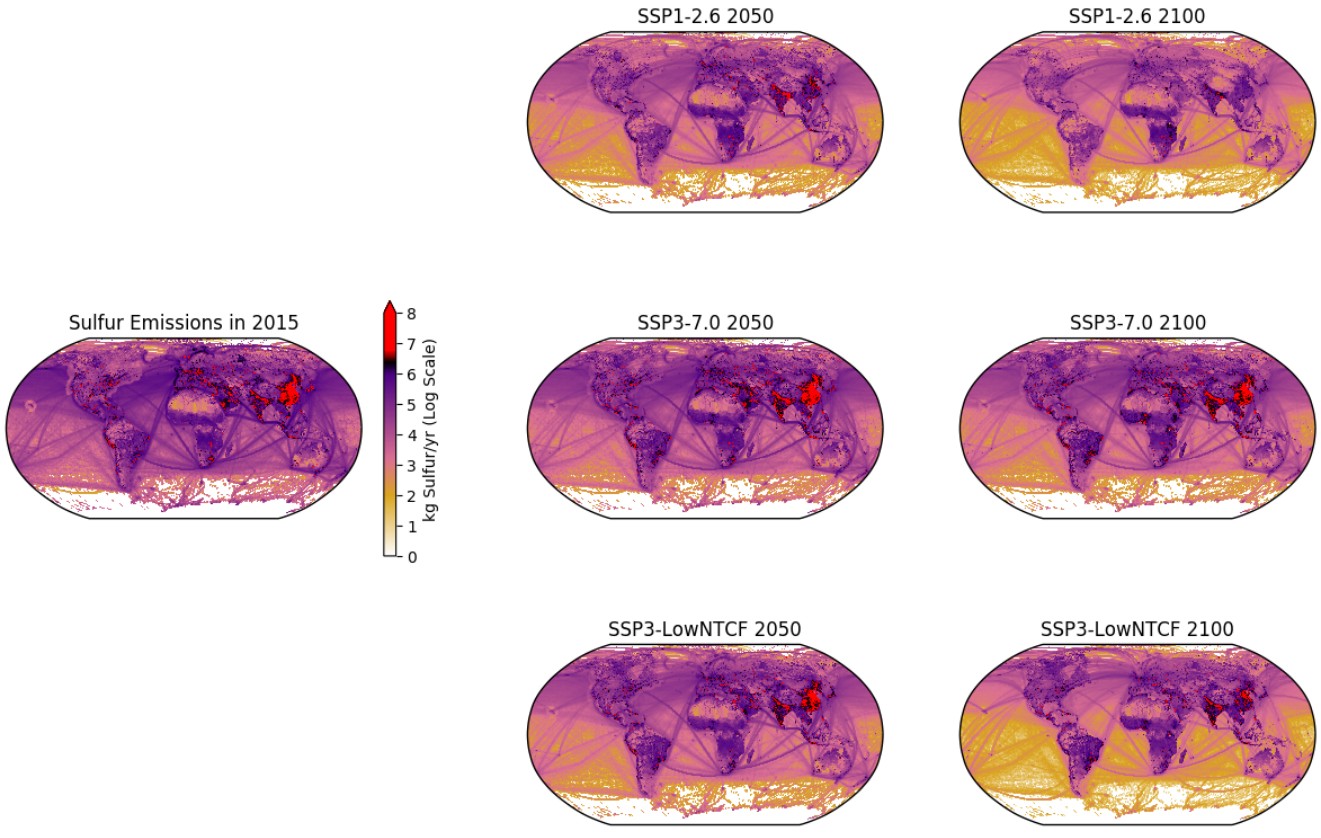

**Figure 10.** Downscaled and gridded emissions of sulfur at present and in 2050 and 2100 for SSP1-2.6, SSP3-7.0, and SSP3-LowNTCF.

Five additional scenarios were analyzed in order to enrich the possible studies of physical and climate impact modeling teams as well as support the scientific goals of specific MIPs. For example, SSP3-7.0 and SSP4-3.4 fill gaps in the available EOC forcing targets provided in the RCPs (7.0 and 3.4 $\mathrm{W\,m^{-2}}$, respectively). A variant of SSP3-7.0, SSP3-LowNTCF (which has similar $CO_2$ trajectories but massively different $CH_4$ trajectories, for example), is also provided to assess the role of near term

5 climate forcers in the context of AerChemMIP. It furthermore serves as an additional EOC forcing target available to climate teams, reaching ~6.3 $\mathrm{W\,m^{-2}}$. Additionally, SSP1-1.9, which sees forcing peak in 2030 at ~2.8 $\mathrm{W\,m^{-2}}$ before declining to 1.9 $\mathrm{W\,m^{-2}}$ by EOC resulting in EOC temperature increases well below 2°C, is the lowest forcing scenario assessed in ScenarioMIP and provides insights for policy-relevant analyses in the context of the Paris Agreement. Finally, an 'overshoot' scenario, SSP5-3.4-OS, is presented in order to study the effects of delayed climate action on long-term climate and related

10 impacts. This scenario, which follows SSP5-8.5 until 2040, sees forcing peak in 2050 at ~4.5 $\mathrm{W\,m^{-2}}$ before stringent climate action occurs resulting in EOC forcing at ~3.1 $\mathrm{W\,m^{-2}}$. These additional scenarios achieve the overall goal of providing both a variety of statistically different EOC climate outcomes as well as enhanced policy and scientific relevance of potential studies.





These emission data are now being used in a variety of multi-model climate model projections (e.g., Fiedler et al. (2018)), including ScenarioMIP, in order to study a number of scientific questions, such as investigations of the role of uncertainty in future forcing trajectories, the effect of forcing peaking and its uncertain timing, and climate outcomes beyond the end of the century. Identifying sources of uncertainties is a critical component of the larger exercise of CMIP6. As such, it is important that

scientists using these datasets for further model input and analysis take care when assessing the uncertainty not only between scenarios but between model results for a certain scenario. While each scenario is presented by a single model in ScenarioMIP, models have also provided a wider range of results as part of the SSP process.

A multi-model dispersion[3] analysis is discussed in SI Section G in order to provide further insight into the robustness of results of emissions trajectories across models for specific forcing targets. Notably, we observe large disagreement between

models for F-gas trajectories (>100% dispersion by EOC in certain cases); thus uncertainty for these species can be considered large by climate modeling teams. We further observe small but non-negligible EOC dispersion (>20%) for certain aerosol emissions species, including $CO$, $NH_3$, $OC$, and sulfur. In general, dispersion between models of GHG species increases as EOC RF decreases as the wide array of mitigation options chosen to meet these lower climate targets can vary across models. The importance of this measure of uncertainty is also scenario dependent. For example, models in general report low emissions

in SSP1 and high emissions in SSP3; thus, the impact of dispersion may have a higher relevance to climate modelers in SSP3 than SSP1.

The ability for other IAM teams to generate and compare results with ScenarioMIP scenarios is also of considerable importance in conjunction with CMIP6 and, after its completion, for further scientific discovery and interpretation of results. As such, we have striven to make openly available all of the tools used in this exercise. The harmonization tool used in this

study, *aneris*, is provided as open-source software on Github as is the downscaling and gridding methodology. Documentation for both is provided to users online. Such efforts and standardizations not only make the efforts of ScenarioMIP robust and reproducible, but also can prove useful for future exercises integrating a variety of complex models.

*Code and data availability.*  The harmonization tool used in this study, *aneris*, is available at https://github.com/iiasa/aneris and documentation for using the tool is available at http://software.ene.iiasa.ac.at/aneris/. Similarly, the downscaling tool used is available at https://github.

com/iiasa/emissions_downscaling and its documentation can be found at https://github.com/iiasa/emissions_downscaling/wiki. Model data, both unharmonized and harmonized is publicly available at the SSP database (https://tntcat.iiasa.ac.at/SspDb) while gridded data is available via the ESGF Input4MIPs data repository (https://esgf-node.llnl.gov/projects/input4mips/).

---

[3]Dispersion here is defined as the coefficient of variation of model results. The coefficient of variation is defined here as the ratio of the standard deviation to mean (absolute value) of a given population of data. See further discussion in SI Section G.





## Appendix A: SSP Drivers

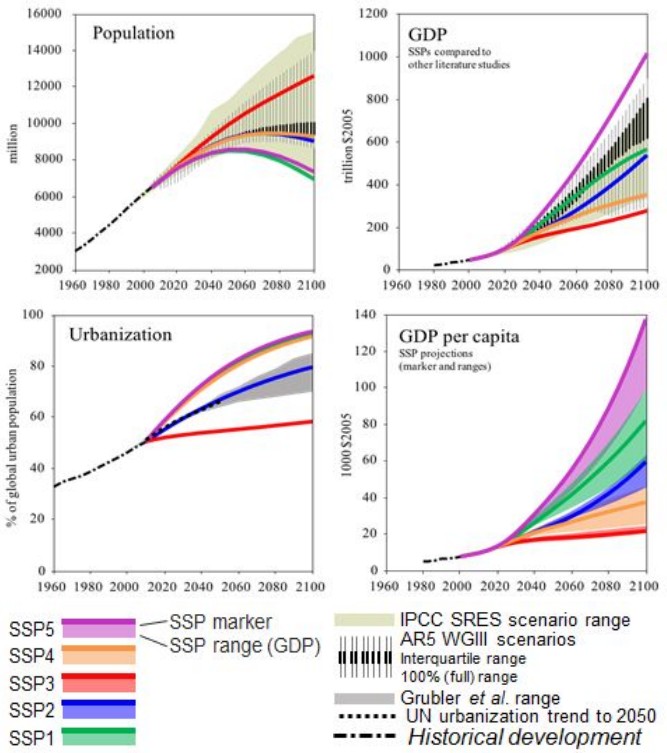

**Figure A1.** The primary socioeconomic assumptions associated with each SSP, including population (KC and Lutz, 2014), urbanization (Jiang and O'Neill, 2015), and GDP (Dellink et al., 2015). The figure is adapted from Riahi et al. (2017) with permission from the authors.

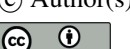



## Appendix B: Supplementary Tables

**Table B1.** The sectoral mapping used to aggregate historical data to a common sectoral definition.

| CEDS Sectors | ScenarioMIP Sectors |
|---:|:---|
| 1A1a_Electricity-public | Energy Sector |
| 1A1a_Electricity-autoproducer | Energy Sector |
| 1A1a_Heat-production | Energy Sector |
| 1A1bc_Other-transformation | Energy Sector |
| 1A2a_Ind-Comb-Iron-steel | Industrial Sector |
| 1A2b_Ind-Comb-Non-ferrous-metals | Industrial Sector |
| 1A2c_Ind-Comb-Chemicals | Industrial Sector |
| 1A2d_Ind-Comb-Pulp-paper | Industrial Sector |
| 1A2e_Ind-Comb-Food-tobacco | Industrial Sector |
| 1A2f_Ind-Comb-Non-metalic-minerals | Industrial Sector |
| 1A2g_Ind-Comb-Construction | Industrial Sector |
| 1A2g_Ind-Comb-transpequip | Industrial Sector |
| 1A2g_Ind-Comb-machinery | Industrial Sector |
| 1A2g_Ind-Comb-mining-quarrying | Industrial Sector |
| 1A2g_Ind-Comb-wood-products | Industrial Sector |
| 1A2g_Ind-Comb-textile-leather | Industrial Sector |
| 1A2g_Ind-Comb-other | Industrial Sector |
| 1A3ai_International-aviation | Aircraft |
| 1A3aii_Domestic-aviation | Aircraft |
| 1A3b_Road | Transportation Sector |
| 1A3c_Rail | Transportation Sector |
| 1A3di_International-shipping | International shipping |
| 1A3dii_Domestic-navigation | Transportation Sector |
| 1A3eii_Other-transp | Transportation Sector |
| 1A4a_Commercial-institutional | Residential Commercial Other |
| 1A4b_Residential | Residential Commercial Other |
| 1A4c_Agriculture-forestry-fishing | Residential Commercial Other |
| 1A5_Other-unspecified | Industrial Sector |





| CEDS Sectors | ScenarioMIP Sectors |
|---|---|
| 1B1_Fugitive-solid-fuels | Energy Sector |
| 1B2_Fugitive-petr-and-gas | Energy Sector |
| 1B2d_Fugitive-other-energy | Energy Sector |
| 2A1_Cement-production | Industrial Sector |
| 2A2_Lime-production | Industrial Sector |
| 2A6_Other-minerals | Industrial Sector |
| 2B_Chemical-industry | Industrial Sector |
| 2C_Metal-production | Industrial Sector |
| 2D_Degreasing-Cleaning | Solvents Production and Application |
| 2D3_Other-product-use | Solvents Production and Application |
| 2D_Paint-application | Solvents Production and Application |
| 2D3_Chemical-products-manufacture-processing | Solvents Production and Application |
| 2H_Pulp-and-paper-food-beverage-wood | Industrial Sector |
| 2L_Other-process-emissions | Industrial Sector |
| 3B_Manure-management | Agriculture |
| 3D_Soil-emissions | Agriculture |
| 3I_Agriculture-other | Agriculture |
| 3D_Rice-Cultivation | Agriculture |
| 3E_Enteric-fermentation | Agriculture |
| 3F_Agricultural-residue-burning-on-fields | Biomass Burning |
| 11B_Forest-fires | Forest Burning |
| 11B_Grassland-fires | Grassland Burning |
| 11B_Peat-fires | Peat Burning |
| 5A_Solid-waste-disposal | Waste |
| 5E_Other-waste-handling | Waste |
| 5C_Waste-incineration | Waste |
| 6A_Other-in-total | Industrial Sector |
| 5D_Wastewater-handling | Waste |
| 7A_Fossil-fuel-fires | Energy Sector |

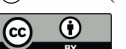


**Table B2.** The sectoral mapping used to aggregate model output data to a common sectoral definition.

| IAM Model Variable | ScenarioMIP Sectors |
|---|---|
| AFOLU\|Agriculture | Agriculture |
| AFOLU\|Biomass Burning | Agricultural Waste Burning |
| AFOLU\|Land\|Forest Burning | Forest Burning |
| AFOLU\|Land\|Grassland Pastures | Grassland Burning |
| AFOLU\|Land\|Grassland Burning | Grassland Burning |
| AFOLU\|Land\|Wetlands | Peat Burning |
| Energy\|Demand\|Industry | Industrial Sector |
| Energy\|Demand\|Other Sector | Industrial Sector |
| Energy\|Demand\|Residential and Commercial and AFOFI | Residential Commercial Other |
| Energy\|Demand\|Transportation\|Aviation | Aircraft |
| Energy\|Demand\|Transportation\|Road Rail and Domestic Shipping | Transportation Sector |
| Energy\|Demand\|Transportation\|Shipping\|International | International Shipping |
| Energy\|Supply | Energy Sector |
| Fossil Fuel Fires | Energy Sector |
| Industrial Processes | Industrial Sector |
| Other | Industrial Sector |
| Product Use\|Solvents | Solvents Production and Application |
| Waste | Waste |

**Table B3.** EOC RF values for unharmonized, harmonized scenario results, and differences between the two. The ScenarioMIP design (O'Neill et al., 2016) states that absolute differences must be within +/- 0.75 $\mathrm{W\,m^{-2}}$, for which all scenarios fall well within the acceptable value.

| Scenario | Unharmonized | Harmonized | Difference | Relative Difference |
|---|---|---|---|---|
| SSP1-2.6 | 2.624 | 2.581 | 0.043 | 1.6% |
| SSP2-4.5 | 4.269 | 4.38 | -0.111 | -2.6% |
| SSP3-Ref | 7.165 | 7.213 | -0.048 | -0.7% |
| SSP4-3.4 | 3.433 | 3.477 | -0.044 | -1.3% |
| SSP4-6.0 | 5.415 | 5.431 | -0.016 | -0.3% |
| SSP5-Ref | 8.698 | 8.424 | 0.274 | 3.2% |

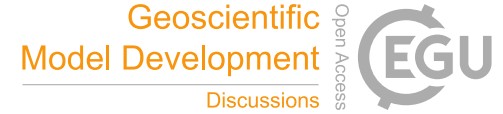

## Appendix C:  Emissions Gridding

Emissions data were mapped to a spatial grid generally following the methodologies described in Hoesly et al. (2018). A brief description is given here, and a fuller discussion of the gridding process will be provided in Feng (2018). For most anthropogenic sectors, emissions at the level of country and aggregate sector are mapped to a 0.5° spatial grid by scaling

the 2010 base-year country-level spatial pattern. For each aggregate sector the spatial pattern of emissions within a country, therefore, does not change over time in the future scenarios, although the spatial pattern of total emissions will change due to changes in the sectoral distribution of emissions. Open-burning emissions from forest fires, grassland burning, and agricultural waste burning on fields are mapped to a spatial grid in the same manner, except that the spatial pattern is taken to be the average from the last 10-years of the historical dataset (e.g., 2005-2014).

International shipping and aircraft emissions are gridded globally such that the global pattern does not change, only the overall emissions magnitude. One other exception occurs for net negative $CO_2$ emissions. Negative $CO_2$ emissions emissions occur in these models when biomass feedstocks are used together with geologic carbon dioxide capture and storage (CCS). In this case, physically, the emissions are taken out of the atmosphere at the locations where biomass is grown, not at the point of energy consumption. In order to avoid large, unphysical, net negative CO2 point source emissions, net negative $CO_2$ quantities

are, therefore, summed globally and mapped to a spatial grid corresponding to 2010 global cropland net primary production (NPP).

## Appendix D:  Global Emissions

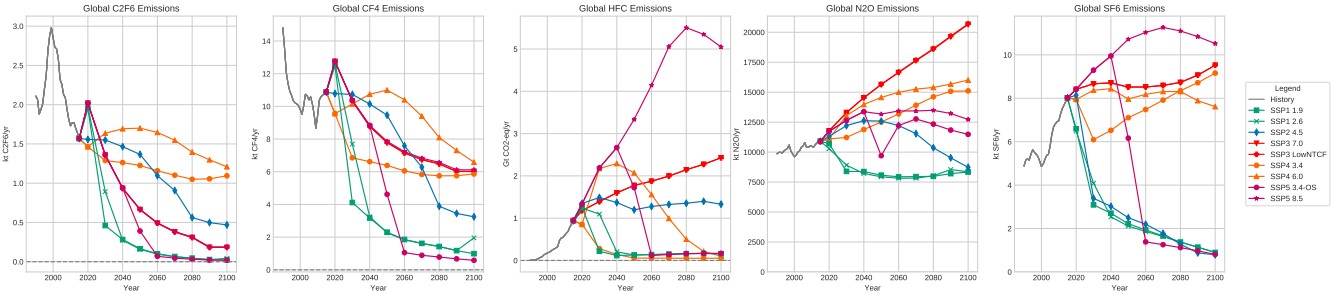

**Figure D1.** Emissions trajectories for all GHGs and all scenarios analyzed in this study.



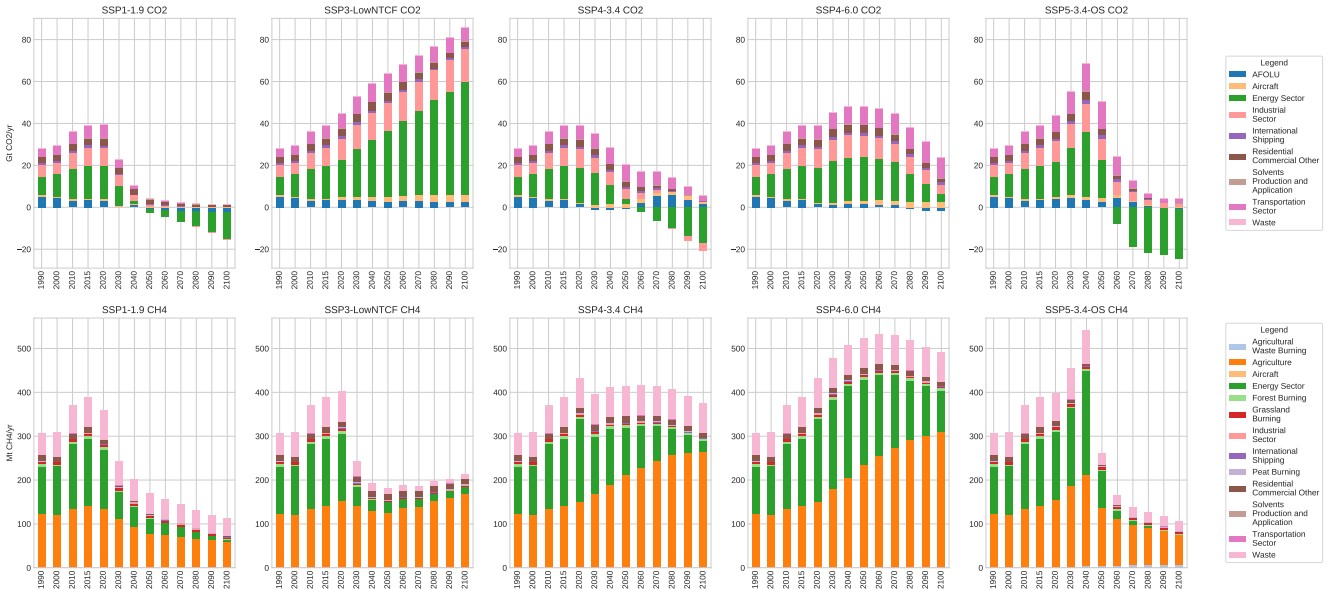

**Figure D2.** Sectoral breakdown for $CO_2$ and $CH_4$ emissions per year for all scenarios analyzed in this study.

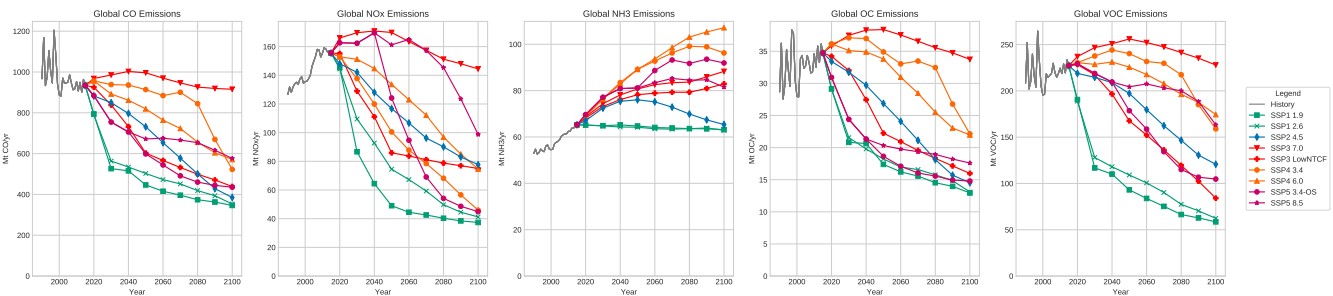

**Figure D3.** Emissions trajectories for all aerosols and all scenarios analyzed in this study.




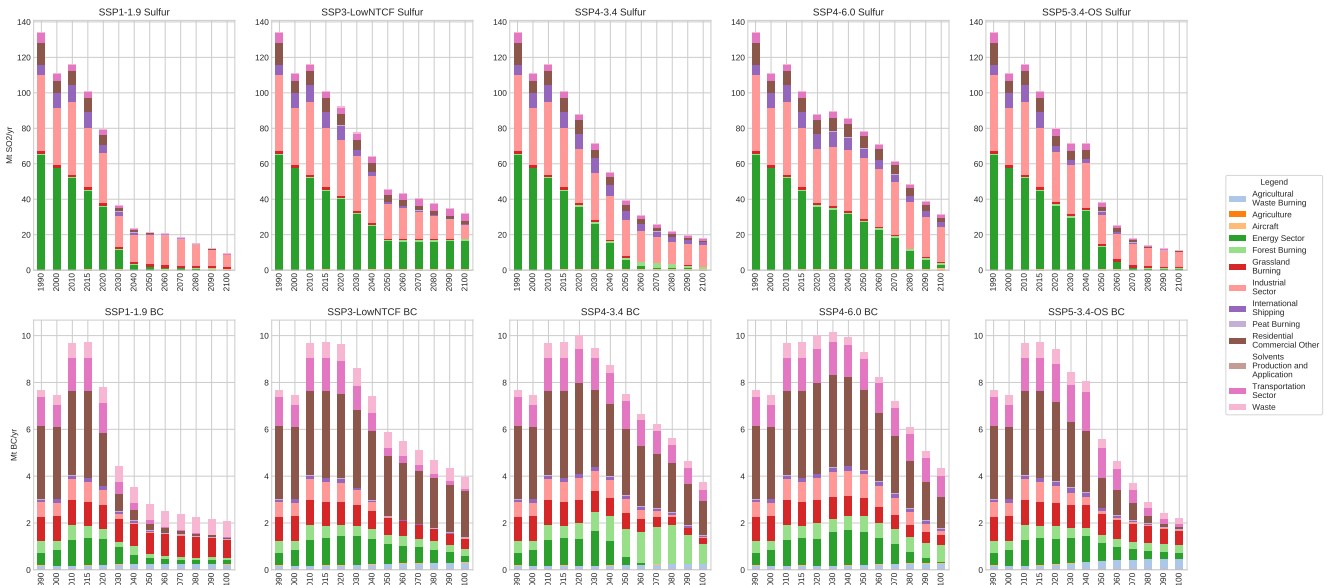

**Figure D4.** Sectoral breakdown for sulfur and BC emissions per year for all scenarios analyzed in this study.

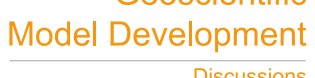
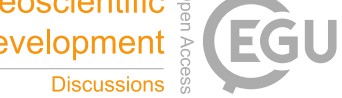


## Appendix E: Harmonization

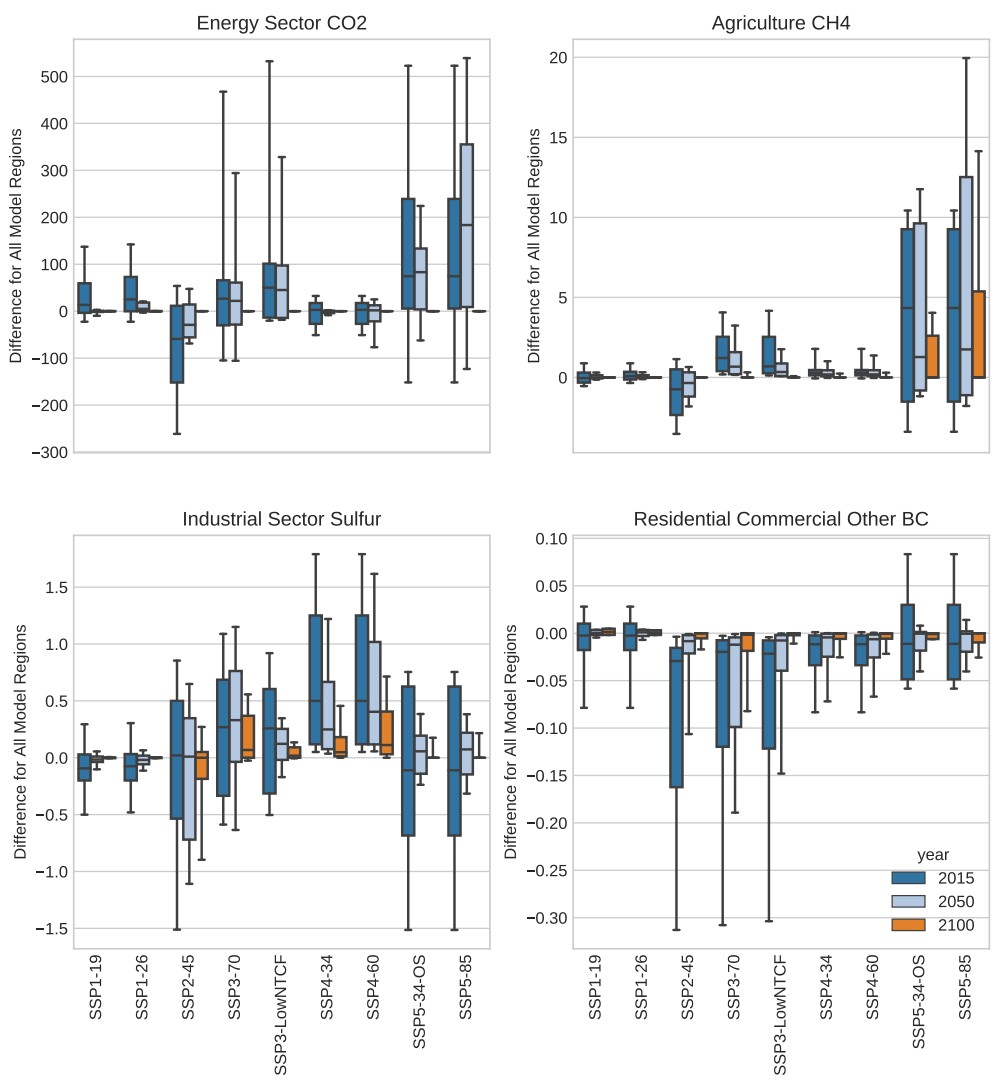

**Figure E1.** The relative difference between harmonized and unharmonized trajectories are shown for the primary sectoral contributor for various emissions species in each scenario. Boxes are comprised of the population of differences for all regions in a given model-scenario combination (see, e.g., Table 3). All box plots show upper and lower quartiles as solid boxes, median values as solid lines, and whiskers extending to 10th and 90th percentiles. In general, the largest deviations are observed in the base year. The spread of values decrease in time across almost all observations, with the convergence to zero or near-zero by EOC.

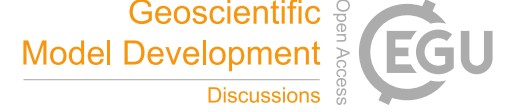

## Appendix F: Regional Emissions

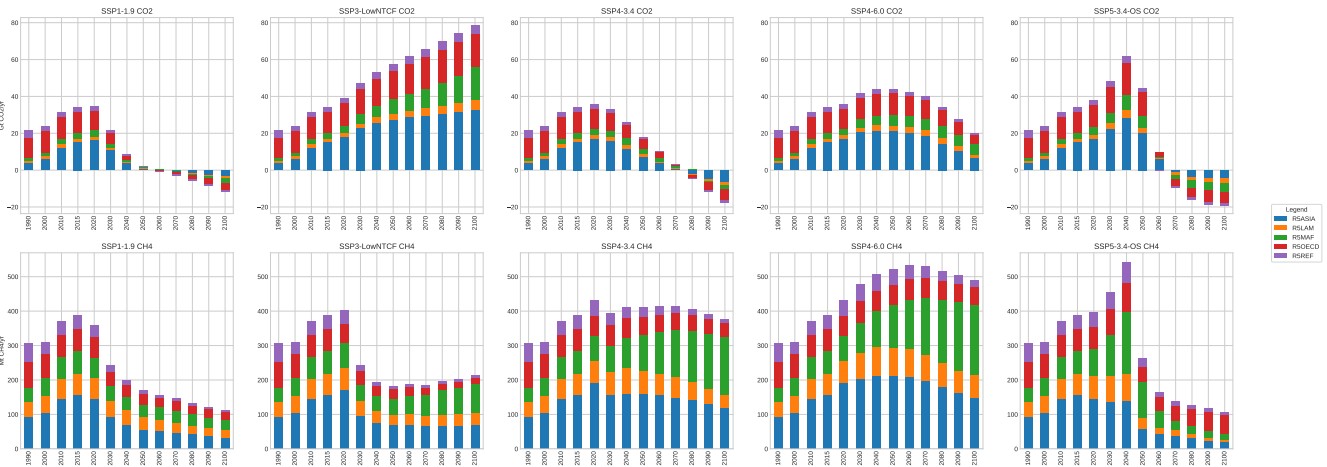

**Figure F1.** Emissions for 5 global regions for all other scenarios analyzed in this study.

## Appendix G: Dispersion Analysis

We here discuss the results of a dispersion analysis measuring the variation of emissions trajectories across models for a given scenario. Dispersion is a measure the spread of model values for a given global emissions value in a given year. It is calculated in this context as the coefficient of variation ($c_v$) shown in Equation G1 which is defined as the ratio of the standard deviation, $\sigma$, to mean, $\mu$, of a given population of data.

$$c_v = \frac{\sigma}{|\mu|} \tag{G1}$$

In order to perform a consistent analysis, we select scenarios for which all participating models provide results: SSP1-2.6, SSP2-4.5, and SSP3-7.0. Scenario data is taken from the available SSP Database at https://tntcat.iiasa.ac.at/SspDb (Riahi et al., 2017). To note, dispersion has a non-zero value in the initial year of analysis due to model results not being harmonized in this dataset. We show the dispersion for GHGs (with aggregated F-gases) in Figure G1, individual F-gases in Figure G2, and aerosols in Figure G3.





**Figure G1.** Dispersion analysis results for GHGs with aggregated F-gases.





**Figure G2.** Dispersion analysis results for individual F-gases.

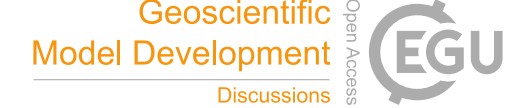



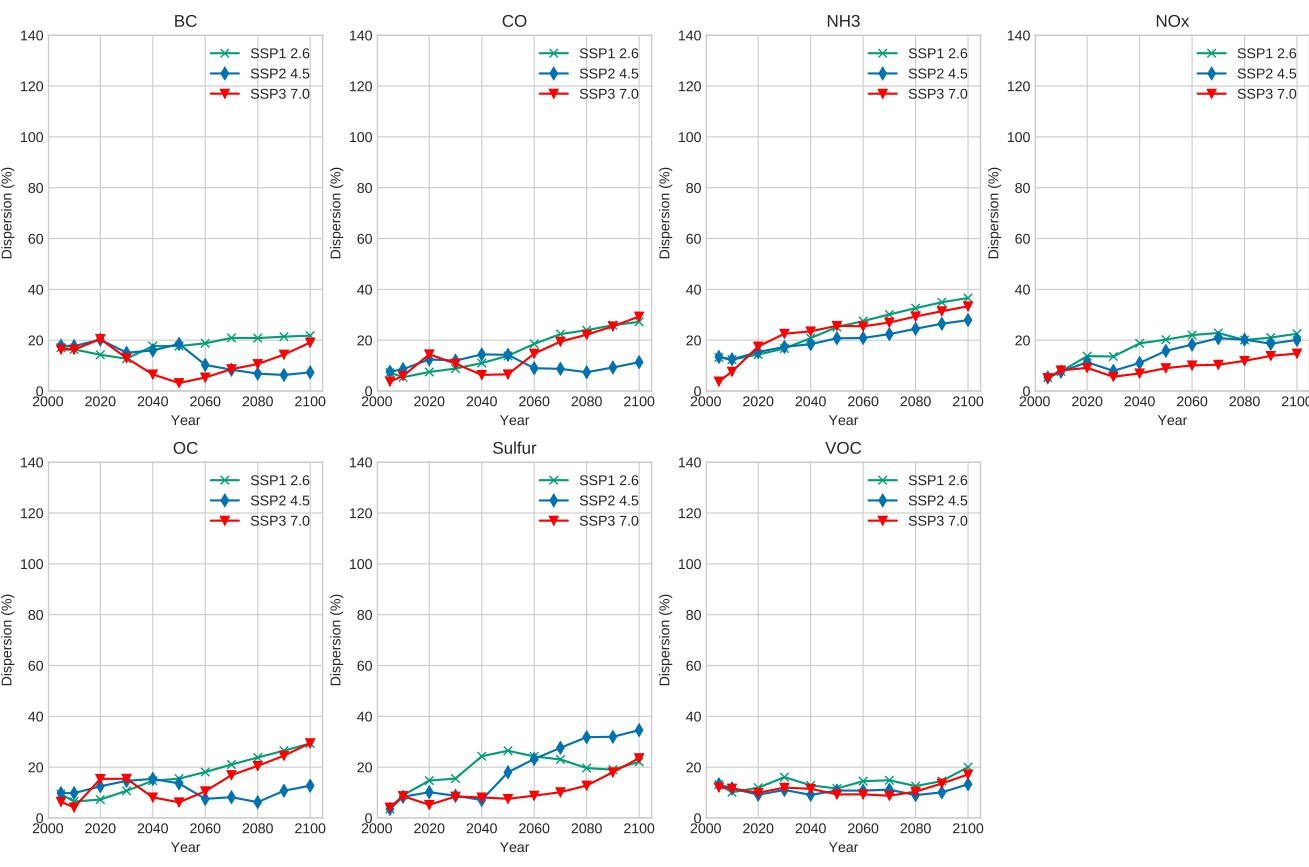

**Figure G3.** Dispersion analysis results for aerosols.

Table G1 shows gas species with the largest values of dispersion. The highest dispersion occurs for F-gases, notably $C_2F_6$, $SF_6$, and HFCs, implying that models generally do not agree on total magnitudes for these gases. $CO_2$ is also observed to have relatively high dispersion in high mitigation scenarios. Finally, aerosol species such as $NH_3$, sulfur, and OC show relative high dispersion values (>30%). In almost every case, magnitudes of emissions with high dispersion decrease substantially with time, thus this measure, while important for understanding sources of error, may result in small total system error in climate models. There are important scenario-species combinations to take account of, however. First, $CO_2$ dispersion in SSP1-2.6 can be of high consequence because this is a scenario with substantial negative emissions at the end of century. Additionally, users of the data should be aware of the dispersion for aerosols in SSP3, as many aerosol species have large EOC magnitudes, thus showing significant variation across models for these species-scenario combinations.





**Table G1.** The dispersion ($c_v$) for the first modeled period and last modeled period for scenarios with maximum model representation. Here we show the 10 highest EOC dispersion values for a given scenario-species combination.

| Scenario | Gas | 2005 | 2100 | Difference | Relative Difference (%) |
|---|---|---|---|---|---|
| SSP1-2.6 | F-gases | 10.96 | 91.31 | 80.34 | 7.33 |
| SSP2-4.5 | F-gases | 10.96 | 89.52 | 78.56 | 7.16 |
| SSP1-2.6 | $CO_2$ | 4.81 | 53.29 | 48.48 | 10.08 |
| SSP2-4.5 | $CO_2$ | 4.80 | 42.63 | 37.83 | 7.89 |
| SSP1-2.6 | $NH_3$ | 13.24 | 36.61 | 23.37 | 1.77 |
| SSP2-4.5 | Sulfur | 3.54 | 34.57 | 31.03 | 8.77 |
| SSP3-7.0 | $NH_3$ | 3.76 | 33.33 | 29.58 | 7.87 |
| SSP3-7.0 | OC | 6.34 | 29.45 | 23.11 | 3.65 |
| SSP1-2.6 | OC | 9.42 | 29.33 | 19.90 | 2.11 |
| SSP3-7.0 | CO | 3.76 | 29.31 | 25.56 | 6.81 |

*Author contributions.* MG, KR, SS, ShF, GL, EK, DV, MV, and DK contributed to the harmonization process and provided extensive data validation efforts. LF led the downscaling effort assisted by SS and MG. KC, JD, StF, OF, MH, TH, PH, JeH, RH, JiH, AP, ES, KT contributed scenario data. All authors contributed to the writing of the paper.

*Competing interests.* The authors have no competing interests.

5 *Acknowledgements.* The authors would like to thank Dr. Joeri Rogelj for his thoughtful comments during the development of this paper. This project has received funding from the European Union's Horizon 2020 research and innovation programme under grant agreement No 641816 (CRESCENDO).



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
