# Peer review of "Global emissions pathways under different socioeconomic scenarios for use in CMIP6: a dataset of harmonized emissions trajectories through the end of the century"

_Geoscientific Model Development, 2018_

## Referee Comment (RC1) · Anonymous Referee #1 · 2 Jan 2019

General Comments

This is a very well written paper. It presents nine newly harmonized and downscaled scenarios for CMIP6 activities that complement scenarios used in CMIP5. The reasoning for the inclusion of the new scenarios are well founded and the analysis is thorough and well organized and presented. The figures complement and are illustrative of the points raised in the text. The harmonization is the result of the application of a methodology using a software package named aneris, which is explained elsewhere in previously published articles. As such, the validity of the harmonization method and

a case study of it have already been subject to peer review.

However, in my opinion, a few points should be addressed before the paper is ready for publication.

One general caveat for acceptance for publication (which led to my "major revisions" request) has to do with the reproducibility of the experiment. On the one hand, the fact that aneris is provided as an open source software package available for download, goes a long way toward reproducibility. However, I was not able to find anywhere in the manuscript a clear link to the data used as input for aneris to perform the harmonization and downscaling. Are these scenarios the same that are available in the public SSP or CMIP database? If so, it should be clearly stated and a link provided. If not, then all the data necessary to replicate the experiment should be made available somewhere and a link provided to it. In my opinion this is required for acceptance of the manuscript for publication.

Additionally, the reference list includes mostly papers authored by the authors themselves. A broader literature review is recommended.

Specific Comments

This article presents a broader application of the harmonization method described in previous publications, but also adds a downscaling process to the scenarios. This downscaling process is partly explained here but the reader is referred to external documentation for further information. In particular, the article Feng (2018) is not yet published, so it is not possible to fully evaluate the validity of the methodology used or of the results obtained. The authors do provide a summary of Feng (2018) in the appendices, but see below for a request for clarification about that.

2.1.3 Region-to-Country Downscaling

Some questions about region-to-country downscaling emerge, particularly about assumptions made. For instance, the use of a linear downscaling method "means that

the fraction of regional emissions in each country stays constant over time" (page 9 line 30). This seems an oversimplification for sectors that may represent a large share of emissions in many countries today (mostly developing countries). However, this may not be true in the future, especially as countries develop and energy use increases. Thus, holding the share of LUC emissions constant over time overestimates their contribution to total emissions, and downplays the potential contribution of energy use in these countries. More importantly, it downplays any mitigation efforts potentially implemented by such countries. Is it reasonable to assume that in SSP1 the shares of Agricultural Waste Burning emissions will remain constant? Why?

This assumption is made without any justification or analysis of its validity and potential impacts on results. The reader is referred to the "downscaling wiki", an online documentation site, but very little explanation is to be found there as well.

Additionally, this linear method may well have different impacts on different GHG species. Aerosols in particular, especially as these are the only species for which the downscaling results are presented in Section 3.4. Surely, agricultural waste burning has an impact on aerosol formation so that the choice of downscaling method in the first step (region to country) is fundamentally important.

Wouldn't it be more appropriate to also use some form of convergence as is done for other sectors? It seems to me this assumption of constant shares for LUC emissions should be fully justified or, preferably, be the subject of sensitivity analysis of some form. This should be further explored, explained and justified. In my view, it should have its own section in the supplementary information. Understandably, there is high uncertainty with these categories of emissions, but that is only more reason to explicitly address it. Also, it would be useful to have a comment by the authors about how this assumption might impact results of the CMIP6 activities using these scenarios as input.

Appendix C: Emissions Gridding

Page 30, line 5: "For each aggregate sector the spatial pattern of emissions within

a country, therefore, does not change over time in the future scenarios, although the spatial pattern of total emissions will change due to changes in the sectoral distribution of emissions."

This sentence is important and should be better formulated. I am having difficulty understanding what stays constant and what changes over time. Maybe the crux of the problem is what is meant by the word "total emissions". Is meant as "global emissions", as contrast with "emissions within a country"?

In my opinion, this section should be expanded to include the points I raised above in my previous comment on Section 2.3.1.

Technical Corrections Page 2, line 32: "where" should be "were" Page 21 line 13: there is no Section 2.4

---

## Referee Comment (RC2) · Anonymous Referee #1 · 3 Jan 2019

This is a correction to my previously submitted review. I overlooked a the "Code and Data Availability" note at the end of the article (i had been looking for footnotes in the body of the article), which led me to believe data was not made available, in which case the experiment would not be reproducible. Because of this I gave a "Major Revision" recommendation and asked for data to be made available. However, soon after submitting the review I saw the note at the end of the article with links to full data. Because my previous review is fully citable, it cannot be edited or retracted, so the editors of GMD advised me to post a correction which I do here.

[Figure]

My recommendation has thus changed to "Accepted subject to minor revisions". Except for the paragraph on making data publicly available, everything else stands. I again commend the authors on an excellent paper. Additionally, because most of the literature cited was penned by the authors themselves, I also recommended a broader literature review, if possible. I do realize that most of the previous work on SSPs was conducted by members of the team of authors of this current article, so it may be challenging to find relevant literature. However, an attempt should be made to be more inclusive so as to avoid excessive self-referencing.

Please accept my sincere apologies for the confusion, and I trust this should not delay publication of the article.

―――――――――――――――――――――

---

## Referee Comment (RC3) · Anonymous Referee #2 · 30 Jan 2019

This manuscript presents a new generation of emission scenarios for their use in CMIP6. This is a very important manuscript for the ESM modeling community because it sets the basis for common model drivers on emission trajectories. The manuscript is very well written. I only have a few minor issues:

- Page 7, lines 10-13. This makes sense only if there's no trend in the data. Did you check whether there are increases or decreases in land burning? Please clarify.

[Figure]

- Eq. 3. If $\dot{I}_c$ is the growth rate, shouldn't the rhs be a sum rather than a multiplication? Please check whether this equation is correct.

- Eq. 5. Please define $c'$. I have problems understanding this equation.

- Section 3.4. The manuscript does a good job describing the methodology for harmonizing the datasets, but it does not describe with the same level of detail the methodology for obtaining the spatial distribution of emissions. I think it's important to add more detail on this methodology to better understand the results presented in section 3.4. In particular, the section describes different values of emissions for the different scenarios in different countries. Since population and GDP change drastically over time and across scenarios, it is very difficult to make sense of sentences such as "...SSP1-2.6, emissions across countries decline dramatically such that by the end of the century, total emissions in China, for example, are equal to that of the USA today". It'd be great if you help the reader by pointing out whether the results in emissions are driven mostly by changes in population, GDP or the other SSP drivers.

- Conclusions. This section is relatively long and reads more like a summary. It can be improved by shortening it, focusing only on the main take home messages of the manuscript, and not summarizing it.

---

## Author Comment (AC1) · 5 Mar 2019

**1 Reviewer 1**

**1.1 General Comments**

Comment

Additionally, the reference list includes mostly papers authored by the authors themselves. A broader literature review is recommended.

Response

We thank the reviewer for their comment regarding the cited literature. The authors of this manuscript do indeed span a number of research teams comprising much of the scenario literature, thus there are inevitably self-references. However, we do agree with the reviewer that our literature review could be more expansive. Therefore, we have increased, where applicable, references to the available literature not including authors on this paper. Notably, we expand our assessment in the user community of emissions scenarios and also with respect to uncertainties in species contribution to radiative forcing.

**1.2 Specific Comments**

Comment

2.1.3 Region-to-Country Downscaling

Some questions about region-to-country downscaling emerge, particularly about assumptions made. For instance, the use of a linear downscaling method "means that

the fraction of regional emissions in each country stays constant over time" (page 9 line 30). This seems an oversimplification for sectors that may represent a large share of emissions in many countries today (mostly developing countries). However, this may not be true in the future, especially as countries develop and energy use increases. Thus, holding the share of LUC emissions constant over time overestimates their contribution to total emissions, and downplays the potential contribution of energy use in these countries. More importantly, it downplays any mitigation efforts potentially implemented by such countries. Is it reasonable to assume that in SSP1 the shares of Agricultural Waste Burning emissions will remain constant? Why?

This assumption is made without any justification or analysis of its validity and potential impacts on results. The reader is referred to the "downscaling wiki", an online documentation site, but very little explanation is to be found there as well.

Response

We apologize for the confusion. Emissions amounts do change over time, it is just the sub-regional distribution amongst countries that is constant for open-burning emissions. We will update the text to clarify that this as:

> means that the fraction of regional emissions in each country stays constant over time. Therefore, the total amount of open-burning emissions allocated to each country will vary over time as economies evolve into the future, following regional trends from the native IAM. However there is no sub-regional change in the within-model-region spatial distribution of land-use related emissions over time. This is in contrast to other anthropogenic emissions, where the IPAT method is used to dynamically downscale to the country level as discussed above.

Comment

Additionally, this linear method may well have different impacts on different GHG species. Aerosols in particular, especially as these are the only species for which the downscaling results are presented in Section 3.4. Surely, agricultural waste burning has an impact on aerosol formation so that the choice of downscaling method in the first step (region to country) is fundamentally important.

Wouldn't it be more appropriate to also use some form of convergence as is done for other sectors? It seems to me this assumption of constant shares for LUC emissions should be fully justified or, preferably, be the subject of sensitivity analysis of some form. This should be further explored, explained and justified. In my view, it should have its own section in the supplementary information. Understandably, there is high uncertainty with these categories of emissions, but that is only more reason to explicitly address it. Also, it would be useful to have a comment by the authors about how this assumption might impact results of the CMIP6 activities using these scenarios as input.

Response

As noted above, open burning emissions are not constant over time. These evolve according to the parent IAM's modeling. The simple IPAT downscaling method was not used because this could lead to illogical results, such as ag waste burning emissions in one country becoming unrealistically large relative to agricultural land area. Certainty improved downscaling methods specific for each type of open-burning emissions are possible. Given that the code has been made available as open source, anyone can implement alternative downscaling methods and test their impact on, for example, aerosol loadings.

Comment

Appendix C: Emissions Gridding

Page 30, line 5: "For each aggregate sector the spatial pattern of emissions within a country, therefore, does not change over time in the future scenarios, although the spatial pattern of total emissions will change due to changes in the sectoral distribution of emissions."

This sentence is important and should be better formulated. I am having difficulty understanding what stays constant and what changes over time. Maybe the crux of the problem is what is meant by the word "total emissions". Is meant as "global emissions", as contrast with "emissions within a country"?

In my opinion, this section should be expanded to include the points I raised above in my previous comment on Section 2.3.1.

Response

This section will be expanded, as follows, to better explain this concept. Note, a companion paper is being submitted that will more fully explain the gridding methodology, results, and implications.

> Emissions data were mapped to a spatial grid generally following the methodologies described in Hoesly et al. (2018). A brief description is given here, and a fuller discussion of the gridding process will be provided in Feng (2019). For most anthropogenic sectors, emissions at the level of country and aggregate sector are mapped to a 0.5° spatial grid by scaling the 2010 base-year country-level spatial pattern. Open-burning emissions from forest fires, grassland burning, and agricultural waste burning on fields

are mapped to a spatial grid in the same manner, except that the spatial pattern is taken to be the average from the last 10-years of the historical dataset (e.g., 2005-2014). For each aggregate gridding sector the spatial pattern of emissions within a country does not change over time in the future scenarios. This means that, for example, the ratio of energy-sector NOx emissions from Shannxi and Beijing provinces in China is constant over time, even though total NOx emissions from China vary over time. Because sectors are mapped to the grid separately, however, total anthropogenic emissions (e.g. sum from all sectors) from any two regions within a country will, in general, not have the same time evolution.

Comment

Technical Corrections Page 2, line 32: "where" should be "were" Page 21 line 13: there is no Section 2.4

Response

We have corrected both typographical errors identified by the reviewer. To note, we have updated all section numbers in Section 2. This was due to an erroneous use of *\subsubsection* instead of *\subsection*.

**2 Reviewer 2**

**2.1 Specific Comments**

Comment

Page 7, lines 10-13. This makes sense only if there's no trend in the data. Did you check whether there are increases or decreases in land burning? Please clarify.

Response

The reviewer here requests clarity on the use of a decadal mean as a harmonization value for land-burning sectors rather than using the value in the most recent historical period as is done for anthropogenic emissions sectors. The variability of emissions species in these sectors is largely due to weather and other climatic variation. In the attached figure we show historical data for the past 25 years normalized to 2005 (the beginning of our data selection window). The harmonization window is shown in grey. While there are some inter-decadal trends, it is hard to observe any specific year-on-year trends. Furthermore, IAM scenarios do not model these year-to-year variations, but only provide a long-term trend. Thus, in order to capture these decadal trends without introducing noise from observations in any specific year, we harmonize to a decadal mean "climatology" as described in the text. We have added the below figure in the SI and referenced it for further clarity to the reader.

Comment

Eq. 3. If $I_c$ is the growth rate, shouldn't the rhs be a sum rather than a multiplication? Please check whether this equation is correct.
[Figure]

Response

Equation 3 is correct, however the growth term was not clearly described in the text. It can more accurately be described as a growth factor. The text has been revised to clarify this, using a less ambiguous notation by introducing the growth factor as $\beta$. Equation 2, provides the annualized amount, as a multiplier, that the initial intensity should grow over time such that intensity will equal the convergence year intensity by the convergence year.

Comment

Eq. 5. Please define $c'$. I have problems understanding this equation.

Response

This equation simply normalizes results to the regional emissions as produced by a given model. $c'$ here is an element of set R (all countries in a region), where the prime superscript is used to differentiate from the index c, which is used in the equation definition. This is common practice in describing elements of sets in order to differentiate the use of the index, thus any further comments here as to if this is still considered confusing would be appreciated (and we can incorporate accordingly). We have provided more clarifying text to describe this operation in the manuscript by explicitly stating it is a normalization.

Comment

Section 3.4. The manuscript does a good job describing the methodology for harmonizing the datasets, but it does not describe with the same level of detail the methodology

for obtaining the spatial distribution of emissions. I think it's important to add more detail on this methodology to better understand the results presented in section 3.4. In particular, the section describes different values of emissions for the different scenarios in different countries. Since population and GDP change drastically over time and across scenarios, it is very difficult to make sense of sentences such as ". . . SSP1-2.6, emissions across countries decline dramatically such that by the end of the century, total emissions in China, for example, are equal to that of the USA today". It'd be great if you help the reader by pointing out whether the results in emissions are driven mostly by changes in population, GDP or the other SSP drivers.

Response

We agree with the reviewer that more detail of the methodology is useful, and a companion paper describing this methodology in much more detail will be submitted to GMD (https://www.geosci-model-dev.net/special_issue590.html) within the week. It is important to note, however, that different models have different regional resolutions. For example, IMAGE is used for SSP1-2.6 and models the USA and China directly. Therefore, there are no downscaling effects associated with the results cited. We strove to discuss the country-specific results where relevant for models in which they are explicitly accounted for. For models which provide country-specific resolution, there are myriad drivers which result in the analyzed emissions trajectories, including not only population, GDP, etc., but also inertia in socioeconomic and energy systems which are endogenously included by the models. Thus, it is hard (if not impossible) to point to a specific subset of these drivers as the primary constituents inducing such changes. In any case, we agree that this can be further clarified, and as such have added the following text to Section 3.4:

Original submission -

CO2 and CH4 are well-mixed climate forcers (Stocker et al., 2013) and thus their spatial variation have a higher impact from a political rather than physical perspective. Aerosols, however, have substantive spatial variability which directly impacts both regional climate forcing via scattering and absorption of solar radiation and cloud formation as well as local and regional air quality. Thus in order to provide climate models with more detailed and meaningful datasets, we downscale emissions trajectories from model regions to individual countries using the methodology described previously in Section 2.4, which are subsequently mapped to spatial grids (Feng, 2018). We here present global maps of two aerosol species with the strongest implications on future warming, i.e., BC in Figure 9 and sulfur in Figure 10. We highlight three cases which have relevant aerosol emissions profiles: SSP1-2.6 which has significantly decreasing emissions over the century, SSP3-7.0 which has the highest aerosol emissions, and SSP3-LowNTCF which has similar socioeconomic drivers as the SSP3 baseline but models the inclusion of policies which seek to limit emission of near-term climate forcing species.

Revision -

CO2 and CH4 are well-mixed climate forcers (Stocker et al., 2013) and thus their spatial variation have a higher impact from a political rather than physical perspective. Aerosols, however, have substantive spatial variability which directly impacts both regional climate forcing via scattering and absorption of solar radiation and cloud formation as well as local and regional air quality. Thus in order to provide climate models with more detailed and meaningful datasets, we downscale emissions trajectories from model regions to individual countries. In most cases, models explicitly represent countries with large shares of emissions (e.g., USA, China, India,

etc.). MESSAGE-GLOBIOM and REMIND-MAGPIE are notable excep-
tions; however, their regional aggregations are such that these important
countries comprise the bulk of emissions in their aggregate regions (e.g.,
the MESSAGE-GLOBIOM North America region comprises the USA and
Canada). For regions constituted by many countries, country-level emis-
sions are driven largely by bulk region emissions and country GDP in each
scenario (per Section 2.4). Afterwards, country-level emissions are subse-
quently mapped to spatial grids (Feng, 2018). We here present global maps
of two aerosol species with the strongest implications on future warming,
i.e., BC in Figure 9 and sulfur in Figure 10. We highlight three cases which
have relevant aerosol emissions profiles: SSP1-2.6 which has significantly
decreasing emissions over the century, SSP3-7.0 which has the highest
aerosol emissions, and SSP3-LowNTCF which has similar socioeconomic
drivers as the SSP3 baseline but models the inclusion of policies which
seek to limit emission of near-term climate forcing species.

Comment

Conclusions. This section is relatively long and reads more like a summary. It can
be improved by shortening it, focusing only on the main take home messages of the
manuscript, and not summarizing it.

Response

We agree with the reviewer that the conclusions section could be more concise. Thus,
we have revised the first few paragraphs in order to encapsulate the primary points
of interest to the reader, while keeping the important points regarding data usage and
availability.

**Supplement:**

---

## Referee Comment (RC4) · Anonymous Referee #3 · 12 Mar 2019

Dear the authors of the manuscript,

This paper describes the key emission data for climate model projection studies. The emission data are being used in the ongoing climate model simulation projects, such as ScenarioMIP. Thus, the significance of the emission data is obvious and a detailed description of the emission data is of great interest to the audience of the GMD, especially the current and future users of the emission data. However, I would suggest some more work to improve the manuscript before being considered for publication.

First of all, some of the key parts in the methodology are supported by unpublished documents that we can't find anywhere, such as Hurtt et al. (2018) for land use, Feng et al. (2018) for gridding, and Meinshausen (2018) for concentrations. This means that we referees are not given enough information to review this manuscript. I don't know why other two referees seemed to be ok with this situation. I should also point out that It's been several months since this manuscript was originally submitted to GMD and we can't still find the documents referred. Since those documents are not available even on the GMD discussion stage, the authors should provide adequate descriptions of those unpublished studies to complete the manuscript as a description paper.

The authors wisely used the references in order not to make the manuscript lengthy. I however feel that some parts of the manuscript needed more description to fully guide the users of the emission dataset (I agree with another referee). As stated in the conclusion section by the authors, the emission data should allow the users to answer scientific questions in the climate studies. In order to do so, this description paper should adequately provide details of the emission data, such as the methodology and underlying data used, and allow the users to figure out science questions that can be (or cannot be) addressed using the emission data and then design studies. For example, the downscaling and emission gridding sections can be greatly improved. The 2.1.3 Region-to-country downscaling section provides the general description. But we don't learn how exactly the authors dealt with the differences in the region definitions among different models. The authors only showed the differences in the region definitions by the numbers of the regions and did not explain well one of the challenging processes in the downscaling. Such information are useful as the users can learn what was done and figure out potential limitations when interpreting the emission data and/or results. Also, the emission gridding section seems to be poor considering the amount of work that the authors must have done. The authors mentioned that the gridding procedure is generally the same as Hoesly et al. (2018) (which is well-written, informative paper in my opinion). Hoesly et al. (2018) provided a great amount of information regarding the data used for emission gridding. Looking at the huge data table in Hoesly et al.

[Figure]

(2018), I would ask what does remain the same in this study and what does not. This might have not been a problem with the Feng emission gridding paper although.

The authors could improve the code data available section, too. In my opinion, the author should provide the list of data they provide from this study with the names of the data. I am suggesting this because I could not find the gridded emission data from the data site listed. Also, if available, the versions of emission data are important to add for traceability.

Sincerely, Referee #3

---

## Author Comment (AC2) · 22 Mar 2019

We thank Referee #3 for their very useful comments. The referee had broadly three main comments regarding first the citation of papers in preparation, second the clarity regarding model regional definitions, and third the clarity of the data availability section. We respond here to these in order.

Regarding the citation of papers in preparation, we fol-lowed the guidance provided in https://www.geoscientific-modeldevelopment.net/for_authors/manuscript_preparation.html, namely:

"Works 'submitted to', 'in preparation', 'in review', or only available as preprint should also be included in the reference list."

We include references to Hurtt et al. (2019) and Meinhausen et al. (2019) primarily for reference to the reader of forthcoming work encompassed in ScenarioMIP which is related to the overall ScenarioMIP experimental design either in parallel to (Hurtt) or derivative of (Meinhausen) this work. We agree with the reviewer that Feng et al. (2019) is of particular importance as a direct methodology and output related to this work. We have been in contact with the author team of that paper and while they expect it to be submitted in the immediate future, we are happy to provide the reviewers with a presubmission version of the manuscript.

Regarding model region information, we have added a footnote to prior work that discusses regional definitions in more detail. In the paper, we state that emissions are harmonized individually for each model to their specified regional definition, and then each regional emission trajectory is downscaled independently using a consistent methodology as outlined in the manuscript.

With respect to the clarity of the data availability section of the paper, we appreciate the reviewer's suggestion and have thus implemented it. Specifically, we have provided additional information with respect to the location of bulk emissions trajectories and describe the filename on ESGF for gridded emissions.